# Observation of two-dimensional Anderson localisation of ultracold atoms

Donald H. White [1,4,6], Thomas A. Haase[1,6], Dylan J. Brown[1,5], Maarten D. Hoogerland [1✉], Mojdeh S. Najafabadi[1,2], John L. Helm[1,2], Christopher Gies[3], Daniel Schumayer [1,2] & David A. W. Hutchinson [1,2✉]

Anderson localisation —the inhibition of wave propagation in disordered media— is a surprising interference phenomenon which is particularly intriguing in two-dimensional (2D) systems. While an ideal, non-interacting 2D system of infinite size is always localised, the localisation length-scale may be too large to be unambiguously observed in an experiment. In this sense, 2D is a marginal dimension between one-dimension, where all states are strongly localised, and three-dimensions, where a well-defined phase transition between localisation and delocalisation exists as the energy is increased. Here, we report the results of an experiment measuring the 2D transport of ultracold atoms between two reservoirs, which are connected by a channel containing pointlike disorder. The design overcomes many of the technical challenges that have hampered observation of localisation in previous works. We experimentally observe exponential localisation in a 2D ultracold atom system.

[1] Dodd-Walls Centre for Photonic and Quantum Technologies, Department of Physics, University of Auckland, Auckland, New Zealand. [2] Dodd-Walls Centre for Photonic and Quantum Technologies, Department of Physics, University of Otago, Dunedin, New Zealand. [3] Institut für Theoretische Physik, Universität Bremen, Bremen, Germany. [4] Present address: Waseda Research Institute for Science and Engineering, Waseda University, Shinjuku, Tokyo, Japan. [5] Present address: Light-Matter Interactions for Quantum Technologies Unit, Okinawa Institute of Science and Technology, Tancha, Onna, Okinawa, Japan. [6] These authors contributed equally: Donald H. White, Thomas A. Haase. ✉email: m.hoogerland@auckland.ac.nz; david.hutchinson@otago.ac.nz

Anderson localisation[1] is a phenomenon resulting from wave interference between multiple propagation paths, and has been observed in a variety of wave systems[2–18]. While it is a single-particle phenomenon, its nature is affected by numerous factors, including interparticle interactions[19,20], dimensionality[21], time-reversal symmetry[22], spin–orbit coupling[23], and the microscopic nature of the disorder[24]. A full understanding of the physics of Anderson localisation demands experimental control of these parameters. Ultracold atoms have proven to be among the cleanest and most controllable of all quantum mechanical systems[25], and have thus provided a natural avenue for modern experiments on Anderson localisation.

Careful experiments in 1D with weakly[26] and non-interacting[27] atoms expanding in a waveguide containing optically generated disorder allowed for unambiguous observation of Anderson localisation. These were followed by experiments demonstrating Anderson localisation in 3D[28,29], and by studies of the metal–insulator transition[30].

In parallel to this, experiments with cold atoms in 2D have shown behaviours characteristic of weak localisation[31–33]. However, unambiguous observation of Anderson localisation in 2D real-space cold atom systems has, to our knowledge, not previously been observed. This has been due to two main challenges. First, the localisation length in 2D depends exponentially on the particle energy[3,34]: for experimentally feasible particle energies, observing localisation requires large systems (>100 μm × 100 μm) even for ultracold atoms. The optically disordered potential landscapes must have high optical resolution over the entire domain, because the scatterer size must be smaller than the atomic de Broglie wavelength (equivalently, the spatial Fourier components of disorder must exceed the majority of atomic momenta). Secondly, while optical speckle patterns provide appropriate disorder for 1D and 3D systems, the statistics of optical speckle are problematic in 2D due to the high classical percolation threshold[35]. Observing Anderson localisation in 2D on reasonable length-scales, therefore, requires relatively strong scattering, and this leads to difficulty in distinguishing localisation effects from classical trapping; low energy particles have the shortest localisation lengths, yet they are also trapped classically by the optical speckle. To this end, Morong and DeMarco[35] suggested the use of randomly positioned point scatterers, which allows for a tuneable percolation threshold based on the amount of disorder, and thus allows for quantum interference effects to be effectively isolated from trapping effects.

In this work we implement point scatterers in a 2D plane by projecting a blue-detuned 532 nm optical pattern shaped by a spatial light modulator (SLM) onto a flat, large-area two-dimensional trap formed from 1064 nm light[36]. The SLM enables any arbitrary potential to be projected onto this plane. We take advantage of this flexibility and project the outline of an additional dumbbell-shaped container consisting of two reservoirs separated by a channel[37,38], with point scatterer disorder located in the channel. Atoms from a $^{87}$Rb Bose–Einstein condensate (BEC) are loaded into the source reservoir, and propagate through the channel into the drain reservoir. The transmissive nature of this experiment has four main advantages compared to traditional expansion experiments with ultracold atoms[39]. Firstly, the fraction of atoms collected in the source and drain reservoirs provides a measurement of the effective resistance of the disordered channel. The measurement of the atom number in a finite reservoir provides a larger signal-to-noise ratio than is accessible with an expansion experiment. Secondly, measuring the atom distribution within the channel enables us to identify the onset of strong localisation as the channel density profile becomes exponential. The two complementary measurements, of the resistance and the channel profile, provide rich information on the transport properties of the disordered channel. Thirdly, the transmissive nature of the experiment allows us to arbitrarily change the length and width of the atom container, and thus to observe the atom transport on length scales both shorter and longer than the localisation length $\xi$. Finally, in a transport experiment the Bose gas is not in thermal equilibrium, which suppresses the formation of a Lifshits glass[40,41] (the mixture of low-energy single-particle localised states could mistakenly be identified as Anderson localisation). With these advantages, we tune between the weak- and strong-localised regimes[42], and observe compelling evidence for Anderson localisation of ultracold atoms in 2D.

## Results

**Evolution of channel density profiles.** The experimental setup consists of ultracold atoms propagating from a source reservoir, through a disordered channel, and into a drain reservoir. The optical setup is illustrated in Fig. 1a, the dumbbell-shaped architecture of the environment is illustrated in Fig. 1b, and the setup is described in detail in Haase et al.[36]. The disorder is characterised by its fill-factor $\eta$, defined as: $\eta = A_{\text{disorder}}/A_{\text{channel}} = n\sigma^2$, where $n$ is the density of scatterers and $\sigma = 1.4$ μm is the effective scatterer width. Equivalently, $\eta$ is the fraction of bright pixels within the channel displayed by the SLM. Note that the classical percolation threshold of point scatterer disorder is negligible for $\eta \lesssim 0.06$ and remains below that of the optical speckle up to $\eta \lesssim 0.35$[35]. In referring to ref. [35], note that our definition of fill-factor differs by a factor of 2, i.e., $\eta = 2nw^2$. We quantify the transport properties of this system in two ways. First, we analyse the long-time behaviour of the atomic density profile within the channel, which allows direct observation of exponential localisation. Second, we measure the temporal behaviour of the source, channel and drain populations ($N_s$, $N_c$, $N_d$). This facilitates the measurement of the transmission coefficient of the channel, which we interpret as a channel "resistance"[37].

We first analyse the long-time behaviour of the system. The signature of Anderson localisation is an exponentially decaying wavefunction, such that the density decays in space with a localisation length of $\xi$ as

$$\rho(x) = \rho_0 e^{-2x/\xi}. \tag{1}$$

After many scattering events, the density of atoms within the disordered channel evolves to exhibit an exponentially decaying profile in an Anderson-localised system. Note that 2D is a special case: although there is a distribution of atomic momenta, and therefore a distribution of localisation lengths, the density profile is expected to be exponential[43]. This is a consequence of the finite (yet possibly large) localisation length for all momenta in 2D.

Figure 2a–c plots the time evolution of the channel density profile for three different fill-factors ($t = 0$ is when the atoms are released from the $CO_2$ laser trap). For weak disorder, we observe a near constant density profile at short evolution times, which evolves to a non-exponential profile for long times. Highly disordered channels ($\eta \geq 0.17$) show distinctly different behaviour. All evolution times over 50 ms indicate an exponential profile. The apparent localisation length, found from the gradient of $\log(\rho(x))$ curve and plotted in Fig. 2d, approaches a quasi-stationary value for long expansion times. The solution of the Gross–Pitaevskii equation shows similar behaviour, superimposed with an oscillation about a constant value. We extract the localisation length measurement from the mean value for expansion times larger than 200 ms and present the result in Fig. 2e. These data indicate that we achieve a localisation length shorter than the channel length of 180 μm for $\eta \gtrsim 0.25$, with a similar threshold observed for the shorter and wider channel

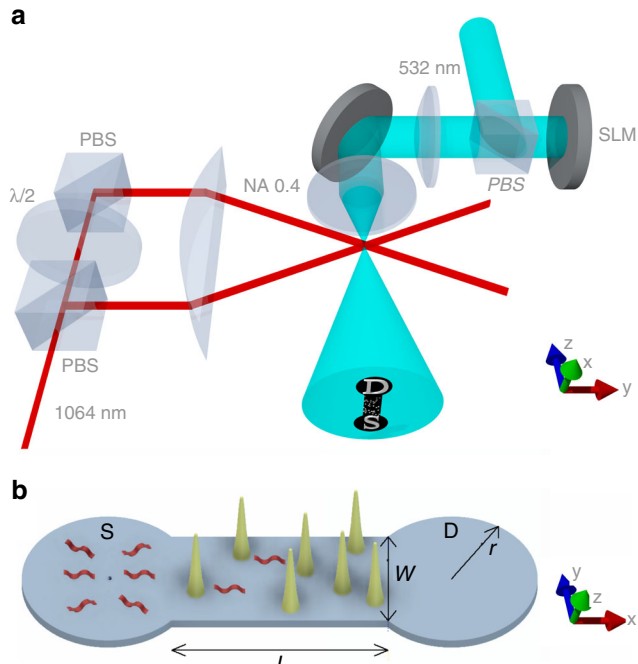

**Fig. 1 Experimental setup. a** The two-dimensional trap is produced by interfering two 1064 nm beams, focused with a 250 mm focal length cylindrical lens. The beams intersect at a relative angle of 6° producing horizontal pancake-like interference fringes in the vacuum chamber where the Bose-Einstein condensate (BEC) is prepared in a crossed-beam $CO_2$ laser optical dipole trap. Atoms from the BEC are loaded into a single light sheet. Simultaneously, a wide and uniform beam of blue-detuned 532 nm light (top right) is reflected from the spatial light modulator (SLM), with a dumbbell-shaped mask applied. Disorder is located within the channel connecting the two reservoirs of the dumbbell. The polarising beamsplitter (PBS) converts the spatial polarisation modulation of the SLM to intensity modulation, which is imaged onto the atomic plane using two lenses. The in-vacuum aspheric lens of numerical aperture 0.42 provides a resolution of 0.9 μm. An example of the dumbbell-shaped combined red and blue optical potentials at the atomic plane are shown in the expanded bottom right bubble. Atoms are loaded into the source (S) reservoir and propagate through the channel into the drain reservoir (D). **b** Atoms are released from the $CO_2$ laser trap at the centre of the source reservoir, and propagate as matter waves (red) into the disordered channel and drain reservoir. The radius $r$, channel length $L$ and channel width $w$ are illustrated. Point disorder within the channel is illustrated as a series of potential hills.

meeting the criterion for strong Anderson localisation. We find a clear relationship showing a reduced localisation length with increasing fill-factor. Numerical simulations give localisation lengths in reasonable quantitative agreement with experiment. We also remark on the slightly stronger localisation observed in the wider 58 μm channel compared to the 43 μm channel: while a fuller investigation of the width-dependence of the localisation length is planned for further study, here we hypothesise that the longer localisation length in the narrower channel is due to finite size effects, associated with the localisation length being significantly longer than the channel width.

We find a difference in equilibration time between theory and experiment in Fig. 2a, though for longer times ($t > 400$ ms) we confirm that the numerical simulations do tend to a near-flat constant density profile (refer to Supplementary Section IV). We attribute differences between the experiment and simulation to effects which are not directly included in the simulation (including finite temperature effects, the smooth disordered potential, and the deviations from flatness in the 2D trap). While

these differences may result in minor deviation between experiment and theory, both point toward Anderson localisation. We also confirm that the simulations in Fig. 2b, c exhibit an exponential channel profile with a quasi-stationary mean localisation length for very long times in the case of $\eta \geq 0.17$, for $t > 400$ ms (refer to Supplementary Section IV).

**Momentum dependence.** Can we attribute the observed exponential density profiles to quantum interference (Anderson localisation)? The alternative interpretation would be classical trapping within the disordered potential for atoms with energies below the percolation threshold. The numerical simulations in Fig. 3a, b show the initial $k$-distribution following the interaction driven expansion from the BEC. The plot in Fig. 3e shows the steady state $k$-distribution within the channel for three different fill-factors. Based on this momentum distribution for $\eta = 0.32$, only 0.8% of atoms within the channel have an energy below the disorder percolation threshold, as calculated according to Morong and DeMarco[35]. This low fraction of classically trapped atoms allows us to be confident that any observed localisation is indeed due to quantum interference.

The momentum distributions, obtained by numerical simulation and illustrated in Fig. 3, provide further insight into the system dynamics. High energy atoms propagate into the drain reservoir. The difference in the drain momentum distributions between zero-disorder and disordered systems (Fig. 3f) shows that the disordered channel acts as an effective energy-filter, preventing low energy atoms ($|k| \lesssim 1.5$ μm$^{-1}$) from propagating into the drain. We interpret the complete inhibition of propagation of low-energy atoms as signifying Anderson localisation. The filtering effect is slightly stronger for $\eta = 0.32$ compared to $\eta = 0.17$. Weakly localised atoms are in an extended state of the system and are able to accumulate in the drain. The momentum distribution in the source (Fig. 3d) is skewed to low energy, because the dwell time within the source reservoir is inversely proportional to $|k|$. The channel (Fig. 3e) contains a mixture of weakly localised high-energy atoms in an extended state across the dumbbell, and strongly localised low-energy atoms. We note that the channel clearly contains a larger number of very low energy atoms ($|k| < 1$ μm$^{-1}$) when disorder is present, indicating that these low-energy atoms are localised within the channel.

In Fig. 3c, we plot the localisation length expected according to

$$\xi(|k|) = \ell_s e^{\pi|k|\ell_{tr}/2}, \qquad (2)$$

where $\ell_s \approx \sigma/\sqrt{\eta}$ is the scattering mean free path and $\ell_{tr} = \Lambda(|k|\sigma)\ell_s$ is the transport mean free path[3,34] (refer to Supplementary Section XI). The curve in Fig. 3c, based on estimates of the mean free path within the system, predicts localisation lengths which are shorter than the system size for $|k| \lesssim 0.55$ μm$^{-1}$. We emphasise that this estimate should be considered in the context of the sensitive exponential dependence of the parameters, the specific microscopic details of the disorder, and the finite size of the system, which are not included in the general estimate of Eq. (2). We note that previous theoretical investigations using point disorder obtained a sub-exponential dependence of $\xi(|k|)$[35], with localisation lengths on the order of 100 μm expected up to $|k| = 6$ μm$^{-1}$, in a system with mean free paths of $\ell_s \approx \ell_{tr} \approx 2$ μm, similar to our experiment. We conclude that our experimental regime is within the bounds set by the established theory, in which Anderson localisation can be expected to be observed. At the same time, classical trapping plays a negligible role in the dynamics within the channel.

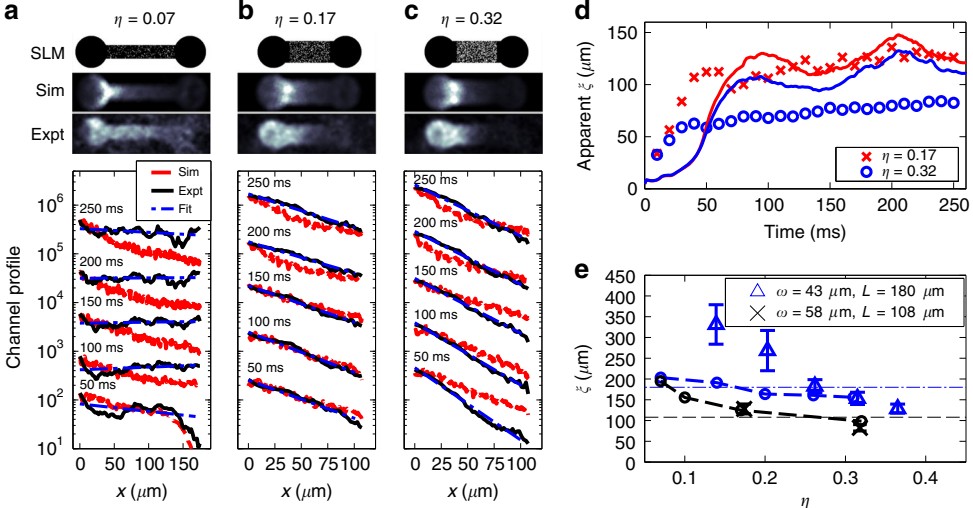

**Fig. 2 Observation of exponential channel density profiles. a–c** The top images in each column show the mask applied to the spatial light modulator (white indicates bright pixels). The second row of images shows the density obtained from Gross–Pitaevskii simulations after 250 ms. The third row of images show an average of three experimental absorption images after 250 ms of evolution, each with different disorder realisations. The channel density profiles show semi-logarithmic snapshots of the channel density (in units of atoms per 2.1 μm camera pixel length), at times (50, 100, 150, 200, and 250) ms of time evolution, with the density integrated across the $y$-direction. Each increasing-time snapshot is offset for clarity by a factor of 10. Profiles are overlaid with an exponential fit to the data in blue, and with the density profiles from the GPE simulation in red. Details of the geometry are: **a** $\eta = 0.07$, $(r, L, w) = (43, 180, 36)$ μm; **b** $\eta = 0.17$, $(r, L, w) = (43, 108, 58)$ μm; **c** $\eta = 0.32$, $(r, L, w) = (43, 108, 58)$ μm. **d** The apparent localisation length is found at each time evolution from the weighted exponential fit to the channel profile for two values of $\eta$, with $(r, L, w) = (43, 108, 58)$ μm. Results from GPE simulations are shown as solid lines. **e** The localisation length is found as an average of apparent localisation lengths for times 210–250 ms, for two channel geometries. Numerical simulation data is also plotted (joined circles). Errorbars show standard deviations obtained from three trials with different disorder realisations. Dotted lines indicate the channel lengths of 180 and 108 μm of the two different geometries. Note that the experimental data for $w = 58$ μm is shown for $\eta = 0.17$ and $\eta = 0.32$ only.

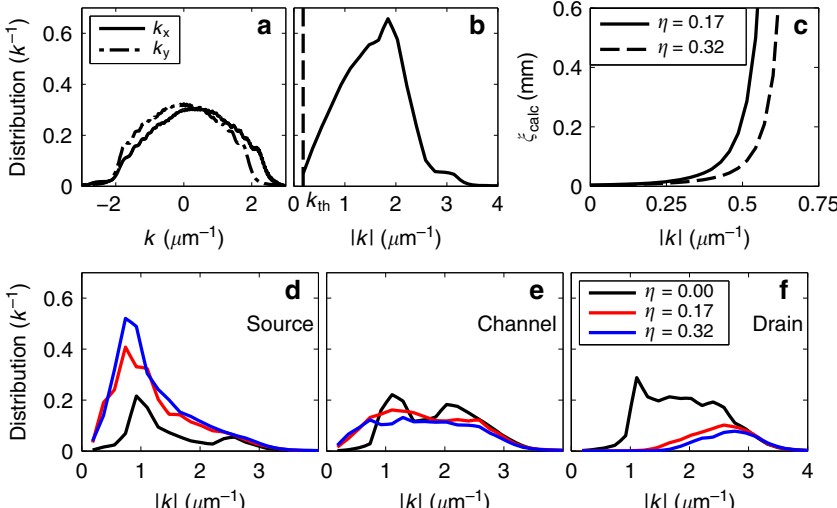

**Fig. 3 Momentum distribution of atoms and the link to localisation length. a** Numerical simulation showing the initial momentum distribution in the $x$ and $y$ directions, following the interaction-driven expansion after release from the harmonic trap, and prior to entering the channel (40 ms of expansion). **b** Distribution of initial absolute momentum value, with a mean of approximately 1.6 μm$^{-1}$. The classical trapping threshold is indicated as the dashed line annotated by $k_{th}$. **c** Theoretical localisation length as a function of $|k|$, as calculated from Kuhn et al. for two fill-factors[34]. Following 250 ms of expansion in the numerical simulation, the $k$-distribution in the three dumbbell regions of a $L = 108$ μm dumbbell is plotted for three fill-factors for **d** the source reservoir; **e** the channel; and **f** the drain reservoir. The scale is chosen such that the integral of each distribution reflects the sum of atoms in that region of the dumbbell.

**Effect of interactions**. The role which interactions play in Anderson localisation has been richly debated in the literature[3,19,20,44–46]. This experiment is conducted with $1.6 \times 10^4$ atoms, resulting in an average density of ~1 atom/μm$^2$. With this

level of atomic density, the interaction energy is significantly lower than either the mean kinetic energy or the disordered potential depth. The experiment is conducted in a regime of weak repulsive interaction, and our numerical simulations indicate that

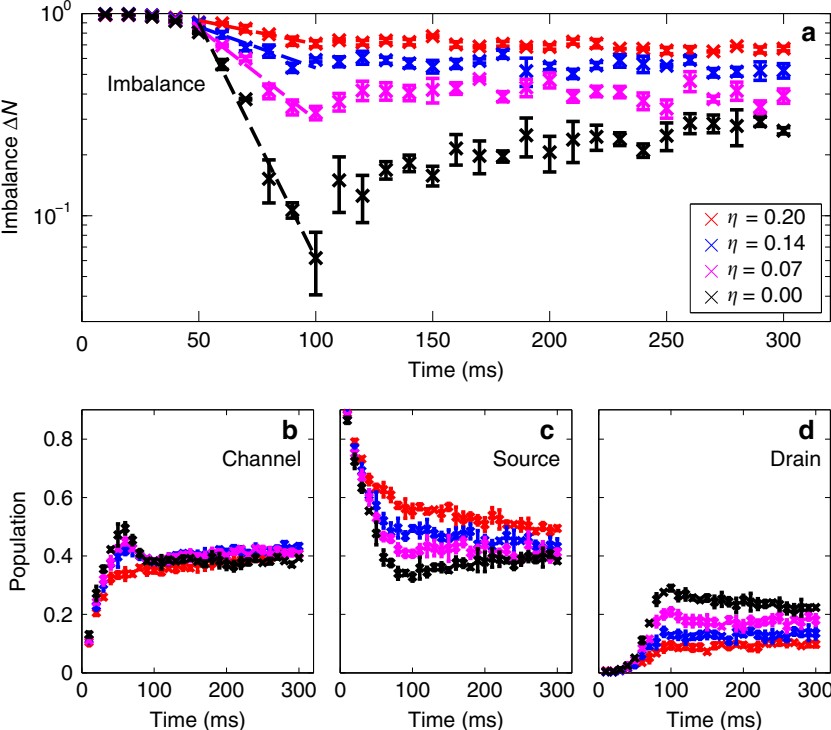

**Fig. 4 Temporal evolution of atom populations. a** The number imbalance $\Delta N$ vs. time for four different fill-factors, with $(r, L, w) = (43, 162, 36)\,\mu m$. Plots are overlaid with the linear fits to the semilogarithmic plot used to calculate the resistance via Eq. (3). **b** Evolution of the channel population. **c** Evolution of the source reservoir population. **d** Evolution of the drain reservoir population. The errorbars show standard deviations in the data, over three disorder realisations.

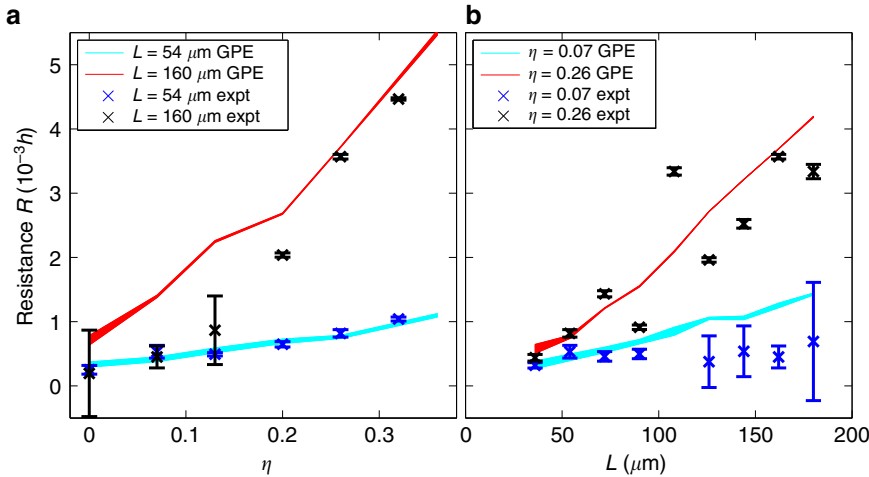

**Fig. 5 Channel resistance measurement. a** The resistance as a function of fill-factor, for two channel lengths, in units of Planck's constant, with $(r, w) = (43, 36)\,\mu m$. **b** The resistance a function of length, for two fill-factors, with $(r, w) = (43, 36)\,\mu m$. Results are overlaid with GPE simulations, with the shaded region indicating one standard-deviation on the simulation value. The errorbars show standard deviations in the data, over three disorder realisations.

the observed localisation length would be unchanged within error for the non-interacting case. Based on our numerical analysis, we estimate that interaction strengths more than five times the experimental interaction would be required to significantly alter the observed density profiles (refer to Supplementary Section X).

**Atomtronic analysis**. For a second complementary analysis, we treat the system as an "atomtronic" circuit[37] and describe the transport in terms of the atomic current flowing between two reservoirs of capacitance $C$ but impeded by a channel resistance $R$. The atomic current is defined by the number imbalance between the source and drain reservoirs: $\Delta N = (N_s - N_d)/(N_d + N_s)$.

Esslinger and co-workers suggested[47] the phenomenological relation

$$\frac{d\Delta N}{dt} = -\frac{\Delta N}{RC}. \tag{3}$$

The data, in Fig. 4, show the evolution of $\Delta N$ for varying fill-factors and for three timescales. In the ballistic period atoms transport across the channel and arrive at the second reservoir. In this first period, the imbalance remains unity due to an empty second reservoir. Following the ballistic time, there is a period of ~40 ms during which the imbalance reduces at its greatest rate. This initial transfer rate is greatest for zero disorder. In this

period, we find an approximately linear relation between $\log(\Delta N)$ and time, supporting the RC circuit model Eq. (3), and we use this transport period to measure the channel resistance. Finally, the system moves into a third regime of transport, in which the number imbalance approaches a steady state, nonzero value for finite $\eta$. This steady-state behaviour is supported by our numerical GPE simulations. We interpret this nonzero steady state number imbalance to be a consequence of a combination of Anderson localisation, and enhanced reflection into the source reservoir due to weak localisation. We note that disorder with low fill-factor (e.g., $\eta = 0.07$) significantly reduces transport, as evidenced by the nonzero steady-state imbalance, although exponential localisation is not observed in this case (cf. Fig. 2a). The reduction in transport is a significant observation, due to the nearzero percolation threshold of $\eta = 0.07$ disorder[35]. In Fig. 4b–d, we show the populations of the channel, source, and drain reservoirs as a function of time. While the reservoir populations show a dependence on $\eta$, the channel population is largely independent of $\eta$ and approaches a steady state. Weak localisation effects within the channel lead to an enhanced reflection coefficient into the source reservoir, and we estimate the mean dwell time within the channel to be 110 ms[48]. This estimated dwell time is largely independent of the details of the disorder and coincides with the population equilibration time. We also note that the steady-state channel population agrees with the relative area of the channel with respect to the whole dumbbell.

The resistance in units of Planck's constant, $h$, is plotted in Fig. 5 for a range of fill-factors and lengths. In this system, $hC = 19$ s (refer to Supplementary Section VII). We observe a stronger dependence of the resistance on fill-factor for longer channel lengths; likewise, we observe a stronger dependence of the resistance on channel length for stronger disorder. While we expect the resistance to be exponential in the channel length in the strongly localised regime[21], here we observe a slower dependence within the accessible experimental parameters. Figure 3f shows that the atoms in the drain have significantly higher energy than the atoms in the channel or source, and we conclude that the main contribution to the resistance measurement comes from atoms with very large localisation lengths and energies larger than the mean energy. We note close agreement between the experimental data and numerical simulations for the resistance measurements.

## Discussion

In conclusion, in combining a highly tuneable experimental platform with full numerical GPE simulations, we have provided compelling evidence for Anderson localisation in a two-dimensional ultracold atom system. For atoms traversing a disordered 2D potential of point scatterers in a regime of weak atomic interaction, we demonstrate clear signatures of exponential localisation. We observe temporally stable exponential channel profiles for $\eta \geq 0.17$. The logarithm of these profiles are linear and do not change significantly for $t > 100$ ms. We have shown for our system that this localisation cannot be explained by classical trapping within the channel. The supporting numerical simulations show that transport of low energy atoms is almost totally inhibited by the disordered channel. We therefore interpret profiles with localisation lengths shorter than the channel length to signify Anderson localisation in 2D.

Through measurements of the localisation length, we have demonstrated that the transport may be tuned via the disorder fill-factor from a regime of ballistic, to diffusive, and then to strongly localised transport with $\xi < L$. The dumbbell-shaped architecture enabled two complementary analyses, allowing

measurements of the channel resistance, together with the in-channel density evolution. The channel resistance indicates that while atoms traversing weak disorder ($\eta = 0.07$) do not exhibit Anderson localisation, the transport is significantly reduced from the zero-disorder case, despite the near-zero percolation threshold. All experimental observations are supported by zero-temperature Gross–Pitaevskii calculations, and the experimental conditions are within the bounds for observation of localisation set by the established theory. The numerical simulations reproduce all signatures observed in the experiment, differing only in equilibration time. The simulations provide additional insight into the role of interactions and the momentum distributions at different fill factors, corroborating the experimental evidence, and providing strong support that Anderson localisation is the suitable interpretation of the exponential density profiles and of the reduced transport. These results provide a springboard for studying localisation and the causes of delocalisation in 2D systems with a quantum-simulator-like device.

## Methods

**Experiment**. A BEC of $^{87}$Rb atoms is initially prepared in a crossed-beam $CO_2$ laser optical dipole trap and $\sim 1.6 \times 10^4$ atoms in the $|F = 1, m_F = -1\rangle$ state are loaded into a large-area quasi-2D trap, as illustrated in Fig. 1. The trap is created by interfering two elliptical beams (1.8 mm-to-8 mm height-to-width ratio), each of 5.0 W of 1064 nm light at an angle of 6°. The resulting light sheets are vertically spaced by 8 μm, while the initial diameter of the three-dimensional BEC is $\sim 2$ μm. This allows the $\sim 5$ nK atoms to load into a single light sheet, with characteristic trap frequencies of $\nu_x = \nu_y = 1$ Hz, $\nu_z = 800$ Hz. This geometry produces a nearly flat potential in the horizontal dimensions, allowing near-ballistic transport with the exception of a weak long-period in-trap interference fringe (refer to Supplementary Section I).

A custom optical potential, produced with an image of a $1280 \times 768$ pixel Holoeye LC-R 720 SLM is then projected onto the working plane. The image is generated with blue-detuned 532 nm light, and imaged with an in-vacuum aspheric lens of numerical aperture 0.42 to give a resolution of 0.9 μm. A single SLM pixel has dimensions 20 μm × 20 μm, which with a magnification of 0.036 translates to a dimension of 0.72 μm × 0.72 μm in the image plane. The setup allows any custom potential to be drawn, and we image a dumbbell-shaped boundary of two reservoirs of radius $r$ separated in the $x$ direction and linked by a channel of length $L$ and width $w$. The channel contains customisable, point-like, optical disorder, produced by images of randomly located blocks of $2 \times 2$ SLM pixels. In the image plane these manifest as approximately Gaussian potential hills of full-width-at-half-maximum $\sigma = 1.4$ μm and 5 nK amplitude.

Atoms are loaded at the centre of the source reservoir and propagate through the channel into the drain reservoir for an expansion time $t$ after the $CO_2$ laser crossed-beam trap is released. The atoms initially expand due to repulsive atom-atom interactions, acquiring kinetic energy and a mean wavenumber of $k \approx 1.6$ μm$^{-1}$. The disorder correlation length is approximately one quarter of the de Broglie wavelength, giving the wave scattering properties which allow for Anderson localisation, especially for atoms with energies lower than the mean energy.

Once the atoms have been loaded into the 2D trap, they are left to expand through the channel into the second reservoir. We impart a weak slope to the trap, giving a gravitational acceleration of $\sim 0.002$ m/s$^2$ along the longitudinal direction and thereby atoms acquire $\sim 0.6 k_B T$ of kinetic energy while crossing a 150 μm channel. This linear potential assists the transport by compensating for a weak fringing barrier (refer to Supplementary Section I) at the opening of the source reservoir and it is analogous to a weak voltage applied to an electronic thin film in order to obtain a resistance measurement. For sufficiently weak bias, Anderson localisation is expected to be maintained[49]. Data acquisition is performed by capturing a series of absorption images, with imaging resolution of 8 μm, at different expansion times within the dumbbell in steps of 10 ms up to 250 ms. Example absorption images are shown in the "Expt" panels of Fig. 2a–c. For each fill-factor the experiment is repeated three times, each time with a different disorder realisation to perform configurational averaging.

**Theory**. Our experimental observations are complemented with a systematic numerical analysis in order to understand the experimental findings in more detail and to support their interpretation. On a fundamental level Anderson localisation is a single-particle phenomenon, therefore, its appearance in a quantum system can be captured by a one-body Schrödinger equation with a potential term, $V_{\text{trap}}(\mathbf{r})$, corresponding to the confinement and to the 2D static disorder. However, in the experiment some weak interaction, $V_{\text{int}}(\mathbf{r})$, is still present between the particles. The interplay between interactions and localisation is a topic of rigorous debate[19]. We note several theoretical studies suggest that localisation is maintained in the presence of weak interactions in 1D[45,50,51], as well as experiments in the many-

body localised regime of strong interactions[44,52]. In the presence of interactions the dynamics are governed by the GPE[53]

$$ih\frac{\partial\psi}{\partial t} = \left[-\frac{h^2}{2m}\nabla^2_{2D} + V_{trap}(\mathbf{r}) + V_{int}(\mathbf{r})\right]\psi, \qquad (4)$$

which we solve using the adaptive, fourth-order Runge–Kutta–Fehlberg method[54]. Our numerical simulations give access to all experimentally observed quantities and we present them alongside of the experimental measurements for comparison. A further advantage of the numerical simulations is that they allow us to switch off the interactions following the initial expansion, allowing us to analyse the effect of interactions on Anderson localisation (refer to Supplementary Section X).

## Correspondence

Correspondence and requests for materials related to the experiment should be addressed to Maarten D. Hoogerland (m.hoogerland@auckland.ac.nz), while queries regarding the theoretical investigation should be directed to David A.W. Hutchinson (david.hutchinson@otago.ac.nz).

## Data availability

All data presented in this publication is available upon request.

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

## Acknowledgements

The authors would like to thank A.V.H. McPhail and I. Herrera for laboratory assistance, and S.S. Shamailov for detailed discussions. D.H.W. thanks L. Sanchez-Palencia and D. Delande for discussions. C.G would like to thank the German Academic Exchange Service (DAAD) for financial support during his stay at the University of Otago. This work was supported by the Marsden Fund, grant number UOA1330, administered by the Royal Society of New Zealand.

## Author contributions

M.D.H. and D.H.W. planned the research. T.A.H., D.H.W., and D.J.B. constructed the experiment. T.A.H. performed the measurements, with D.J.B. and D.H.W. providing the assistance. D.J.B. and D.H.W. carried out the data analysis. J.H. wrote the Gross–Pitaevskii code. M.S.N. and D.S. ran and analysed the simulations, and together with C.G and D.A.W.H. formed the theoretical underpinning. M.D.H. and D.A.W.H. supervised the experimental and theoretical work, respectively. All authors discussed the research and contributed to the paper.

## Competing Interests

The authors declare no competing interests.
