## [Peer Review File · Nature Communications]

Reviewers' comments:

Reviewer #1 (Remarks to the Author):

This manuscript describes a joint experimental and theoretical work where Anderson localization was observed in the 2D transport of coherent ultracold atoms between source and drain reservoirs in the presence of controlled obstructions. Briefly Bose-Einstein-condensed (BEC) atoms were released into the source well of a dumbbell-shaped potential and allowed to propagate through a channel filled with point-like obstructions into the drain well. The data collected included channel density profiles and channel resistance measurements. Theoretical analyses of the experiment were carried out by solving the 2D Gross-Pitaevskii equation (GPE).

I think that the work presented in this manuscript is technically sound. The observation of 2D Anderson localization in ultracold-atom transport is a new result as far as I know and these results should be of interest to researchers in the field of atomtronics. Thus I believe that this manuscript deserves to be published in Nature Communications.

I do have one concern that I think must be addressed by the authors before publication. In the last paragraph the authors state that the most conclusive evidence for the presence of Anderson localization are the linear channel density profiles (plotted on a logarithmic scale) for a fill factor above $\eta > 0.17$. Then later they say that all experimental observations are supported quantitatively by GPE calculations. However, the comparison between theory and experiment for the channel densities displayed in Figs. 2 (a), (b), and (c) do not exhibit "quantitative" agreement. It is true that both theory and experiment show linear profiles for $\eta > 0.17$, these profiles do have some differences. Furthermore, for $\eta = 0.007$, I would not call the comparison "quantitative" agreement.

I think that the authors might want to revisit the statement about quantitative agreement. They should also add a short (a few sentences) discussion about the differences between theory and experiment. I hasten to add that I believe the evidence presented here for Anderson localization is compelling. I am worried that the stated characterization of the theory/experiment comparison is not borne when looking at Fig. 2.

If the authors add the short discussion described above, I recommend that this manuscript be published in Nature Communications.

Reviewer #2 (Remarks to the Author):

Manuscript NCOMMS-19-38098-T reports a combined experimental-theoretical investigation of Anderson localization in a disordered atomic system in reduced dimensionality. The authors explore the transmission of ultracold, weakly-interacting atoms along a ribbon-like channel subjected to binary disorder realized with an optical potential. The goal is to observe the elusive Anderson localization in a 2D environment, an important goal of potential strong interest for a wide community. As the authors discuss in the introduction, the phenomenon has so far escaped observation because of two main issues that are peculiar of the 2D environment: a strong energy

dependence of the localization length, which can be easily affected by finite-size effects; the presence, for the widely-used speckle disorder, of a very high percolation threshold that might mix Anderson localization with classical trapping.

To solve such issues, the authors employ two innovative methods. First, they realize a ribbon-like, long channel between two particle reservoirs, to ensure a large system size at least along one direction. A similar configuration was previously employed by a group at ETH to study the behaviour of disordered Fermi gases (ref. 52). The present work follows a different approach, since they do not explore regimes of weak imbalance of the population of the two reservoirs but prepare initially all the atoms in one reservoir. Second, they realize a disordered potential consisting of point scatterers, to reduce the impact of the percolation threshold, as was studied theoretically by the authors of ref. 35. The advantage is however clear only for weak disorder, while for strong disorder the percolation threshold becomes again relevant as in the speckles.

The authors study various observables: 1) The quasi-stationary density profile that develops in the channel at long times; 2) The time dependence of the atom number in the reservoirs; 3) The resistance of the channel, which comes from an elaboration of the atom-number dynamics, following ref. 52. The experimental measurements are compared to numerical simulations based on the Gross-Pitaevskii equation, finding in a general a good agreement.

The authors claim observation of Anderson localization in 2D on the basis of the behaviour of the density profiles in the channel (observable 1), to my understanding with the following arguments: For sufficiently strong disorder (large filling factor of point scatterers), the density profiles become exponential and show little variation over time, suggesting the onset of strong localization. The average localization length is shorter than the channel length (although there is a rather strong dependence on the channel width that the authors are not able to explain). The system is not in thermal equilibrium, so the occupation of strongly localized states in the Lifshits tail is suppressed. The experimental data agrees with the results of the numerical simulations (although the agreement does not seem good for the data at weak disorder shown in Fig.2).

From the study of observables 2-3, the authors deduce instead that there is an initial stage of transport when the first atoms traverse the channel, followed by a later stage in which the channel/source/drain populations stabilize to quasi-stationary values, with a finite imbalance between the source and drain populations, and approximately constant channel population. From the initial evolution of the drain and source populations, they deduce a channel resistance, which apparently increases by increasing the disorder strength and the channel length. Since the increase of the resistance with length is only roughly linear and not exponential, as instead expected for strong localization, the authors conclude that there must be a relevant contribution of energy states with localization length longer than the channel. The long-time stationary populations are instead attributed to a combination of Anderson localization and classical trapping below the percolation threshold. In a couple of points, the authors note that the transport is significantly affected by a weak disorder, in a regime where Anderson localization does not seem to take place.

In my opinion, the study by White and coworkers presents interesting novel experimental data on transport in the presence of disorder but it does not provide a convincing evidence of the elusive

Anderson localization in 2D. I see various weaknesses in the authors' arguments, various contradictions in the discussion, and in general a lack of quantitative comparison of the results to the assessed theory of disordered systems. All that makes me think that this work represents just an initial, non-conclusive study of the phenomenon, and that much more experimental work and more comparison with the assessed theory must be done to observe Anderson localization in 2D. So, I cannot give a recommendation towards acceptance of the present manuscript.

Here are my main points of criticism:

- 1) As the authors note, the study in ref.35 found that the classical percolation threshold of point-scatterers disorder stays below that of speckle disorder for filling factors smaller than 0.35. However, this does not mean that such threshold is zero. Ref. 35 shows indeed that the threshold is a relevant fraction of the disorder energy for the typical values of the filling factors for which the present work claims observation of Anderson localization based on the density profiles (0.17-0.32). Then, how can the authors be sure that such exponential localization is not due to classical localization? While there is no discussion about this potential issue for the data in Fig.2, later the authors introduce the possibility of classical trapping for the data in Fig.3, for the same range of disorder strengths. No quantitative analysis of such important issue is however provided. Note that the versatile experimental setup realized by the authors would allow them to change the point scatterers disorder into speckle disorder, to check the impact of the different percolation thresholds.
- 2) The data in Fig.2 shows a non-negligible effect of the channel width on the localization length. The authors note the effect, but they do not offer an explanation and propose to study it in the future. One notes also that the average localization length along the channel is larger than the channel widths. So, I would say that this work does not explore a truly 2D problem.
- 3) Regarding the transport measurements in Fig. 3, the authors claim that the steady-state imbalance at long times is due to Anderson localization (as well as to classical trapping). One notes, however, that there is a finite imbalance also for the disorder-free case. Such observation seems to invalidate the argument about Anderson localization. How do the authors justify such imbalance? Might it be due to some spurious effect linked to the fact that they explore only regimes of strong initial imbalance, hence strongly out-of-equilibrium systems (differently from ref. 52)?
- 4) The discussion contains an apparent contradiction. From the data in Fig.2, the authors deduce that they have observed Anderson localization ("the most conclusive experimental and numerical sign of the onset of Anderson localisation is the exponential channel profiles"). However, they interpret differently the data the data in Fig.3 as the result of both Anderson localization and classical trapping ("We interpret this non-zero steady state number imbalance to be a consequence of a combination of Anderson localisation, and classical trapping for atoms below the percolation threshold").
- 5) The experiment-theory agreement is in general good, except for the data at weak disorder in Fig.2. There, the simulations find apparently an exponential profile also in the absence of localization. How can then the authors conclude that an exponential profile must be taken as a solid proof of Anderson localization?

6) The experiments are conducted with a Rb sample, with fixed scattering length. This implies that the system is interacting, so in principle one might not speak of Anderson localization, which is a single-particle phenomenon. Of course, one might be in a regime of weak interactions, which do not affect substantially the dynamics of the system. I guess that it would have been relatively easy to check the potential role of the interactions at least at the numerical level, with a simulation of a non-interacting system. No discussion on this potential issue is however offered.

7) In the supplemental material, the authors present and discuss additional data comparing the density profile of a disordered system with that of a periodic lattice of scatterers. The comparison suggests that an exponential profile is present only for the disordered case, while the regular lattice presents a more complex profile, with an initial slower decay. I find this an important point, that might support the arguments in favour of Anderson localization. However, the comparison is limited to the analysis of the density profiles (it is not even clear whether the data in Fig.5 represent just a single realization of the disorder) and no transport data is shown or discussed.

8) The authors tend to present the general agreement between experiment and simulations as a supporting proof of Anderson localization (see for example the conclusions). I disagree with such point of view, and I think that only a quantitative comparison with the established theory (e.g. ref. 35) could support the arguments in favour of Anderson localization.

Reviewer #3 (Remarks to the Author):

In the manuscript, the authors study two-dimensional (2D) Anderson localization (AL) with ultracold bosonic atoms. Using a spatial light modulator (SLM) they design a trap, which consists of a source and a drain interconnected by a channel. Using the SLM they manage to create a disorder potential of randomly placed Gaussians of sufficiently small width (point-like scatterers) avoiding in this way many limitations previously faced using speckle potentials. By monitoring the particle density in the channel and transport properties through the channel the authors claim to see unambiguously Anderson localization in 2D.

The experimental setup is innovative and has the potential to address many interesting questions in connection to AL. At present stage, however, several important issues need to be addressed before I can recommend the paper for publication.

1) The authors say that investigating the interplay between AL and interaction is interesting but do not seem to consider it relevant for their setting. In fact they claim, "several theoretical studies suggest that localisation is maintained in the presence of weak interactions in 1D", citing two papers. However, several other theoretical studies closely connected to the present manuscript show the opposite: Starting with studies on subdiffusive spreading in discrete nonlinear disordered systems [see e.g. Flach et al. Phys. Rev. Lett. 102, 024101 (2009)] several studies in a continuous speckle potential showed, that AL is influenced by even weak interactions [see e.g. Donsa et al. PRA 96, 043630 (2017); Min et al. Phys. Rev. A 86, 053612 (2012)]. In fact the paper by Donsa et al. used

exactly the parameters of the experiment Billy et al., Nature 453 (2008) (reference [26] in the manuscript) and demonstrated that interactions are important, however, on longer time scales than observed in the experiment. Assuming that the GPE is valid (at least) for coarse-grained observables like the particle density for the Billy et al. experiment I thus also disagree with the statement in the present manuscript that ref [26] unambiguously shows AL of noninteracting atoms in 1D. Even more relevant seems the work by Dujardin et al. Phys. Rev. A 93, 013612 (2016) which is in 1D but otherwise corresponds to the scenario of transport between source and drain. In this paper, it is shown that weak interactions lead to a correction of the localization length while a complete loss of coherence is observed for stronger interactions.

I thus disagree with the author's conclusion that interactions do not play a role in their system. In fact, they miss the opportunity to study the interplay between interactions and Anderson localization following the lines of many previous experimental studies.

2) I find it very surprising that the authors do not mention the value of the chemical potential of the condensate. In fact, the chemical potential and the associated healing length (ξ) is an important energy and length scale, respectively, never discussed in the text. It would allow evaluating how strong interactions are in their experiment. The only hint given is that they are in the Thomas-Fermi regime which points to the fact that interactions are non-negligible. I found it also surprising that the authors deem the thermal wavelength and associated momentum k_B more important than the momentum associated with the healing length $1/\xi$. In fact, the GPE theory they use to compare their experiment with is a $T=0$ theory and no thermal scales enter whatsoever. I, therefore, strongly suggest mentioning the value of the chemical potential and all other scales associated with it.

3) I disagree with the statement that the numerical simulations show quantitative agreement with the experiment. Especially for $\eta=0.32$ the agreement is at best qualitative. I also find it interesting how different the numerically calculated atomic densities look as compared to the absorption images in Fig. 2 a, b, c (why is the source so empty?). Unfortunately, the authors do not give any hints on the origin of these discrepancies. I also do not agree with the statement that the localization length in Fig. 2 d "approaches an asymptotic value for long localization times". In fact, the localization length slowly but steadily increases during the whole observation time again pointing at the importance of interactions.

4) I find the observation of a constant population in the channel quite interesting. At first sight, this might be very counterintuitive since naively one would expect that a larger localization length leads to a smaller population. However, it has been shown for scattering of linear waves in disorder that the dwell time is independent of the disorder strength, see e.g. Pierrat et al. PNAS 111, 17765 (2014). For linear waves the dwell time is associated with the fraction of the wavefunction inside the scattering region. One could directly compare the results of this paper with the so-called Weyl formula. As far as I know investigations of this effect including nonlinearity/interactions is lacking and would provide the current study with a further interesting physics question.

Finally, a few more technical comments.

Main text:

In Fig.1 the image of the trap in the expanded bubble is very hard to see. Also the coordinate system

does not seem to agree with the axis convention in all the following figures.

A plot of the trapping potential as a function of x (a cut through the potential) would be very helpful. At least it should appear in the supplement where the numerical simulations are discussed.

The parameters (r, L, w) are not explained in the main text and one has to guess that it is the radius, the length, and the width of the trap.

Why is the same averaging procedure as for the experimental data (see caption of Fig.2) not applied also to the numerical results?

In the conclusions the sentence appears: "... and demonstrated that the transport may be tuned via disorder fill-factor from a regime of ballistic, to diffusive, and then to strongly localized transport". It should be made more clear that diffusive transport is associated with the period of time from which the resistance is taken.

The sentence in the conclusions "Our work demonstrated and explored the phenomenon..." reads like the repetition of the preceding sentence.

Supplementary material:

Overall, I find Figure 5 interesting/disturbing. While I agree that the randomly placed scatterers show nicer agreement to an exponential density profile, the regularly placed scatterers overall show the same behavior. The authors should explain the reason in more detail since it seems to hint to the excluded absence of classical percolation.

The authors should give also the time step used in their numerical calculations since a standard Runge-Kutta-Fehlberg (please check spelling in your manuscript) for such systems is accurate only for sufficiently small time steps.

In section E in the supplement reference 49 directs to the supplement and thus seems to be obsolete.

In the same section the unit of k_{dB} is given as m .

Overall I find Fig. 7 confusing:

- In the main text it is mentioned that the scatterers are created via 2×2 pixels. Why is $w \geq 20px$ then relevant?
- Why is the data for $w=20px$ shown only for small fill-factors?

Response to referees

We thank the three referees for their comments on our manuscript, titled “Observation of two-dimensional localisation of ultracold atoms”. Their level of detailed engagement with our manuscript is a credit to the peer-review process.

Referee 1 noted that our experimental results, combined with the support of numerical simulations, provide solid evidence for the observation of Anderson localisation in two-dimensions. The referee suggested we add further discussion to the link between simulations and experiments, which we have done in the main text.

Referees 2 and 3 provided two main criticisms of our results. The first concerned whether our observations of steady-state exponential density profiles may be strictly attributed to the interference of quantum paths, or whether classical trapping effects may be mimicking Anderson localisation. As this addresses the central claim of our manuscript – the observation of Anderson localisation – this is an important question and we agree that further analysis will strengthen our manuscript. We re-emphasise that our experiment has been designed to counter the effect of classical trapping to the best of our technological ability, via the implementation of high-resolution point scatterers distributed over a wide region.

In order to conclusively establish that classical trapping does not result in the exponential density profiles which we observe in the experiment and the simulations, we have conducted further analysis based on the existing theoretical work of Morong and DeMarco. Using the numerical simulation tools available, we have analysed the localisation properties based on the momentum distributions of the atoms and we have added a new figure to the manuscript (now Figure 3), which shows the steady-state k -vector distributions within the different regions of the dumbbell. We see that the drain collects atoms which are highly energetic, and we conclude that the resistance measurements in Figure 4 are based on atoms which are not Anderson localised. In contrast, the atoms within the channel are low energy, and we draw conclusions based on the peak value of this distribution. Specifically, we calculate the expected exponential dependence of localisation length on k in Fig. 3(c), and we show that localisation can reasonably be expected to be observed under the experimental conditions. Additionally, we show that only 0.8% of atoms are below the classical percolation threshold. This means that our observation of exponential localisation cannot be attributed to classical trapping. We also conducted some indicative simulations with low levels of temporally-varying phase noise, in order to break time-reversal symmetry. We saw that this resulted in destruction of the exponential profile. Together with the new theoretical analysis, this is strong evidence that our observation of exponential localisation is due to Anderson localisation.

The second main criticism of our work related to the effect of atom-atom interactions in the experiment. Referee 3 correctly notes that the question of the effect of interactions on localisation has been the subject of rich debate for many years, and that with greater emphasis on the role of interactions in our experiment, our results would add significantly to this discussion. We have conducted further numerical simulations in aid of this goal. The results indicate that the main role of interactions in this experiment is to drive the initial expansion of the atoms upon trap release, providing atomic kinetic energy. Once the atoms enter the wide channel, the low atomic density on the order of 1 atom / μm^2 means that the interaction energy is significantly smaller than the other

relevant energy scales in the problem. We perform simulations in which we abruptly set the atomic scattering length to zero following the initial interaction-driven expansion of atoms. These simulations produce results which are quantitatively and qualitatively equivalent to the simulations with the ^{87}Rb equivalent scattering length, indicating that this experiment takes place in a regime of very weak interaction, and we may regard this as an effectively single-particle experiment. We also conducted simulations at larger scattering lengths, to determine a threshold at which interactions would significantly affect localisation. We determine that the interaction energy would need to be at least five times larger to affect the experiment. Our simulations do not show the long-time increase in localisation length indicated by Dujardin et al, PRA 93 013612, for the interaction levels of the experiment.

We here address all points raised by the referees. Our responses to individual points are in indented paragraphs.

Reviewer #1 (Remarks to the Author):

This manuscript describes a joint experimental and theoretical work where Anderson localization was observed in the 2D transport of coherent ultracold atoms between source and drain reservoirs in the presence of controlled obstructions. Briefly Bose-Einstein-condensed (BEC) atoms were released into the source well of a dumbbell-shaped potential and allowed to propagate through a channel filled with point-like obstructions into the drain well. The data collected included channel density profiles and channel resistance measurements. Theoretical analyses of the experiment were carried out by solving the 2D Gross-Pitaevskii equation (GPE).

I think that the work presented in this manuscript is technically sound. The observation of 2D Anderson localization in ultracold-atom transport is a new result as far as I know and these results should be of interest to researchers in the field of atomtronics. Thus I believe that this manuscript deserves to be published in Nature Communications.

I do have one concern that I think must be addressed by the authors before publication. *In the last paragraph the authors state that the most conclusive evidence for the presence of Anderson localization are the linear channel density profiles (plotted on a logarithmic scale) for a fill factor above $\eta > 0.17$. Then later they say that all experimental observations are supported quantitatively by GPE calculations. However, the comparison between theory and experiment for the channel densities displayed in Figs. 2 (a), (b), and (c) do not exhibit "quantitative" agreement. It is true that both theory and experiment show linear profiles for $\eta > 0.17$, these profiles do have some differences. Furthermore, for $\eta = 0.007$, I would not call the comparison "quantitative" agreement. **I think that the authors might want to revisit the statement about quantitative agreement. They should also add a short (a few sentences) discussion about the differences between theory and experiment.** I hasten to add that I believe the evidence presented here for Anderson localization is compelling. I am worried that the stated characterization of the theory/experiment comparison is not borne when looking at Fig. 2.*

- We have conducted new numerical simulations for longer times, and found that in the case of weak disorder ($\eta = 0.07$) for the data of Fig. 2(a), the density profile tends to a constant value after a longer time (400 ms). Thus there is a quantitative disagreement in the equilibration time between theory and experiment. In response to this we have

added a new paragraph: Page 5, Paragraph 2, which discusses the differences between experiment and theory.

If the authors add the short discussion described above, I recommend that this manuscript be published in Nature Communications.

Reviewer #2 (Remarks to the Author):

Manuscript NCOMMS-19-38098-T reports a combined experimental-theoretical investigation of Anderson localization in a disordered atomic system in reduced dimensionality. The authors explore the transmission of ultracold, weakly-interacting atoms along a ribbon-like channel subjected to binary disorder realized with an optical potential. The goal is to observe the elusive Anderson localization in a 2D environment, an important goal of potential strong interest for a wide community. As the authors discuss in the introduction, the phenomenon has so far escaped observation because of two main issues that are peculiar of the 2D environment: a strong energy dependence of the localization length, which can be easily affected by finite-size effects; the presence, for the widely-used speckle disorder, of a very high percolation threshold that might mix Anderson localization with classical trapping.

To solve such issues, the authors employ two innovative methods. First, they realize a ribbon-like, long channel between two particle reservoirs, to ensure a large system size at least along one direction. A similar configuration was previously employed by a group at ETH to study the behaviour of disordered Fermi gases (ref. 52). The present work follows a different approach, since they do not explore regimes of weak imbalance of the population of the two reservoirs but prepare initially all the atoms in one reservoir. Second, they realize a disordered potential consisting of point scatterers, to reduce the impact of the percolation threshold, as was studied theoretically by the authors of ref. 35. The advantage is however clear only for weak disorder, while for strong disorder the percolation threshold becomes again relevant as in the speckles.

The authors study various observables: 1) The quasi-stationary density profile that develops in the channel at long times; 2) The time dependence of the atom number in the reservoirs; 3) The resistance of the channel, which comes from an elaboration of the atom-number dynamics, following ref. 52. The experimental measurements are compared to numerical simulations based on the Gross-Pitaevskii equation, finding in a general a good agreement.

*The authors claim observation of Anderson localization in 2D on the basis of the behaviour of the density profiles in the channel (observable 1), to my understanding with the following arguments: For sufficiently strong disorder (large filling factor of point scatterers), the density profiles become exponential and show little variation over time, suggesting the onset of strong localization. The average localization length is shorter than the channel length (**although there is a rather strong dependence on the channel width that the authors are not able to explain**). The system is not in thermal equilibrium, so the occupation of strongly localized states in the Lifshits tail is suppressed. The experimental data agrees with the results of the numerical simulations (**although the agreement does not seem good for the data at weak disorder shown in Fig.2**).*

- We note a dependence on the width, which is also present in the simulation, but we think it likely that this dependence is a geometric effect associated with reflections from the channel edge. This does not fundamentally alter our results.
- As stated above in the response to Referee 1, the simulation equilibration time is longer than the experimental equilibration time.

*From the study of observables 2-3, the authors deduce instead that there is an initial stage of transport when the first atoms traverse the channel, followed by a later stage in which the channel/source/drain populations stabilize to quasi-stationary values, with a finite imbalance between the source and drain populations, and approximately constant channel population. From the initial evolution of the drain and source populations, they deduce a channel resistance, which apparently increases by increasing the disorder strength and the channel length. Since the increase of the resistance with length is only roughly linear and not exponential, as instead expected for strong localization, the authors conclude that there must be a relevant contribution of energy states with localization length longer than the channel. The long-time stationary populations are **instead attributed to a combination of Anderson localization and classical trapping below the percolation threshold**. In a couple of points, the authors note that the transport is significantly affected by a weak disorder, in a regime where Anderson localization does not seem to take place.*

- In our revised analysis, we have found that only 0.8% of atoms have an energy below the classical percolation threshold. We conclude that classical trapping does not play a significant role in the dynamics. The long-term stationary number imbalance is a result of enhanced reflection into the source reservoir due to weak localisation (quantum interference).

*In my opinion, the study by White and coworkers presents interesting novel experimental data on transport in the presence of disorder but it does not provide a convincing evidence of the elusive Anderson localization in 2D. I see various weaknesses in the authors' arguments, various contradictions in the discussion, **and in general a lack of quantitative comparison of the results to the assessed theory of disordered systems**. All that makes me think that this work represents just an initial, non-conclusive study of the phenomenon, and that much more experimental work and more comparison with the assessed theory must be done to observe Anderson localization in 2D. So, I cannot give a recommendation towards acceptance of the present manuscript.*

- We have expanded our quantitative comparison to established theory, namely in the percolation threshold calculation and the calculation of the expected localisation length in Fig. 3(c).

Here are my main points of criticism:

1) As the authors note, the study in ref.35 found that the classical percolation threshold of point-scatterers disorder stays below that of speckle disorder for filling factors smaller than 0.35. However, this does not mean that such threshold is zero. Ref. 35 shows indeed that the threshold is a relevant fraction of the disorder energy for the typical values of the filling factors for which the present work claims observation of Anderson localization based on the density profiles (0.17-0.32). Then, how can the authors be sure that such exponential localization is not due to classical localization? While there is no discussion about this potential issue for the data in Fig.2, later the authors introduce the

possibility of classical trapping for the data in Fig.3, for the same range of disorder strengths. No quantitative analysis of such important issue is however provided. Note that the versatile experimental setup realized by the authors would allow them to change the point scatterers disorder into speckle disorder, to check the impact of the different percolation thresholds.

- 1.) The new theoretical analysis shows that the percolation threshold is very low with respect to the mean particle energy, and that only 0.8% of atoms are expected to be classically trapped in the disorder. Additionally, new numerical simulations with phase noise indicate that in the absence of interference, the channel density does not evolve to a steady-state exponential profile. We appreciate the referee's comment regarding the experimental creation of an artificial speckle potential for comparison. However, we are currently limited by the technology of the spatial-light modulator, which results in intensity flicker when greyscale values are chosen. This is a topic for future study.

2) The data in Fig.2 shows a non-negligible effect of the channel width on the localization length. The authors note the effect, but they do not offer an explanation and propose to study it in the future. One notes also that the average localization length along the channel is larger than the channel widths. So, I would say that this work does not explore a truly 2D problem.

- 2.) In the supplementary material, we have added experimental data for a broad range of different widths. This data indicates that localisation persists for a large width, suggesting that we are not in a one-dimensional environment (where all states are by definition localised). While we would ideally be in a situation where the localisation length is shorter than both dimensions, we find it difficult to accept that this experimental regime is anything other than 2D. The vertical trapping frequency is 800 times larger than either of the horizontal dimensions; and the width of the channel is over an order of magnitude larger than either the de Broglie wavelength of the atoms or the scatterer dimensions. We note a dependence on the width, which is also present in the simulation, but we think it likely that this dependence is a geometric effect associated with reflections from the channel edge. This does not fundamentally alter our results.

3) Regarding the transport measurements in Fig. 3, the authors claim that the steady-state imbalance at long times is due to Anderson localization (as well as to classical trapping). One notes, however, that there is a finite imbalance also for the disorder-free case. Such observation seems to invalid the argument about Anderson localization. How do the authors justify such imbalance? Might it be due to some spurious effect linked to the fact that they explore only regimes of strong initial imbalance, hence strongly out-of-equilibrium systems (differently from ref. 52)?

- 3.) The long-term stationary number imbalance is a result of a combination of enhanced reflection into the source reservoir due to weak localisation (quantum interference), together with Anderson localisation within the channel. We base this conclusion on the study of Pierrat et al (ref. 56) which found that the dwell time in the channel is largely independent of the details of the disorder, but that the reflection coefficient into the source reservoir is strongly affected by disorder. This is in accordance with the picture of coherent backscattering enhancing reflection into the channel (ref. 32). The finite imbalance in the case of zero disorder is a combination of the finite reflection coefficient at the source/channel boundary, and the "sloshing motion" within the channel. As the referee

suggests, this system does have a strong initial imbalance, and the finite reflection coefficient from the change in impedance between the circular reservoir and the rectangular channel leads to a delayed equilibration time. We also note the “sloshing” motion within the dumbbell, whereby the imbalance goes to zero at time 100 ms, following which atoms are reflected back from the drain into the channel and the source (we note the increase of the absolute source population after 100 ms). The equilibration time to zero imbalance is longer than the experiment time.

4) The discussion contains an apparent contradiction. From the data in Fig.2, the authors deduce that they have observed Anderson localization (“the most conclusive experimental and numerical sign of the onset of Anderson localisation is the exponential channel profiles”). However, they interpret differently the data the data in Fig.3 as the result of both Anderson localization and classical trapping (“We interpret this non-zero steady state number imbalance to be a consequence of a combination of Anderson localisation, and classical trapping for atoms below the percolation threshold”).

- 4.) While the observation of Anderson localisation does not preclude the presence of other physical effects such as classical trapping, it is clear that classical trapping plays a limited role in this experiment (as only 0.8% of atoms are below the percolation threshold). We have amended the manuscript accordingly.

5) The experiment-theory agreement is in general good, except for the data at weak disorder in Fig.2. There, the simulations find apparently an exponential profile also in the absence of localization. How can then the authors conclude that an exponential profile must be taken as a solid proof of Anderson localization?

- 5.) The simulation data for Fig. 2(a) tends to a constant density profile for long times ($t > 500$ ms). Note that the simulated density profiles in Fig. 2(a) are not linear on the sublogarithmic scale, and we do not agree with the referee’s comment that these are “exponential” profiles.

6) The experiments are conducted with a Rb sample, with fixed scattering length. This implies that the system is interacting, so in principle one might not speak of Anderson localization, which is a single-particle phenomenon. Of course, one might be in a regime of weak interactions, which do not affect substantially the dynamics of the system. I guess that it would have been relatively easy to check the potential role of the interactions at least at the numerical level, with a simulation of a non-interacting system. No discussion on this potential issue is however offered.

- 6.) The new numerical simulations, shown in the supplementary material, indicate that we are in a very weakly interacting regime. Additionally, the chemical potential within the entire dumbbell is 0.3 nK, significantly lower than the kinetic energy and the disorder strength.

7) In the supplemental material, the authors present and discuss additional data comparing the density profile of a disordered system with that of a periodic lattice of scatterers. The comparison suggests that an exponential profile is present only for the disordered case, while the regular lattice presents a more complex profile, with an initial slower decay. I find this an important point, that might support the arguments in favour of Anderson localization. However, the comparison is limited

to the analysis of the density profiles (it is not even clear whether the data in Fig.5 represent just a single realization of the disorder) and no transport data is shown or discussed.

- 7.) We have included the resistance data for the regular lattice, as well as numerical simulations of the density plots. The disorder contains multiple disorder realisations, while the regular lattice contains a single realisation (averaged three times)

8) *The authors tend to present the general agreement between experiment and simulations as a supporting proof of Anderson localization (see for example the conclusions). I disagree with such point of view, and I think that only a quantitative comparison with the established theory (e.g. ref. 35) could support the arguments in favour of Anderson localization.*

- 8.) The agreement between numerical simulation and the experiment show that we may trust the simulations to reflect the experimental conditions. Therefore by testing the interaction and phase-noise conditions with the simulations, we can be confident that these tests reflect the experimental reality. Additionally, we have enhanced our quantitative comparison with the established theory, showing that our observations are in accordance with the expected results for our system.

Reviewer #3 (Remarks to the Author):

In the manuscript, the authors study two-dimensional (2D) Anderson localization (AL) with ultracold bosonic atoms. Using a spatial light modulator (SLM) they design a trap, which consists of a source and a drain interconnected by a channel. Using the SLM they manage to create a disorder potential of randomly placed Gaussians of sufficiently small width (point-like scatterers) avoiding in this way many limitations previously faced using speckle potentials. By monitoring the particle density in the channel and transport properties through the channel the authors claim to see unambiguously Anderson localization in 2D.

The experimental setup is innovative and has the potential to address many interesting questions in connection to AL. At present stage, however, several important issues need to be addressed before I can recommend the paper for publication.

1) *The authors say that investigating the interplay between AL and interaction is interesting but do not seem to consider it relevant for their setting. In fact they claim, "several theoretical studies suggest that localisation is maintained in the presence of weak interactions in 1D", citing two papers. However, several other theoretical studies closely connected to the present manuscript show the opposite: Starting with studies on subdiffusive spreading in discrete nonlinear disordered systems [see e.g. Flach et al. Phys. Rev. Lett. 102, 024101 (2009)] several studies in a continuous speckle potential showed, that AL is influenced by even weak interactions [see e.g. Donsa et al. PRA 96, 043630 (2017); Min et al. Phys. Rev. A 86, 053612 (2012)]. In fact the paper by Donsa et al. used exactly the parameters of the experiment Billy et al., Nature 453 (2008) (reference [26] in the manuscript) and demonstrated that interactions are important, however, on longer time scales than observed in the experiment. Assuming that the GPE is valid (at least) for coarse-grained observables like the particle density for the Billy et al. experiment I thus also disagree with the statement in the present manuscript that ref [26] unambiguously shows AL of noninteracting atoms in 1D. Even more relevant seems the work by Dujardin et al. Phys. Rev. A 93, 013612 (2016) which is in 1D but*

otherwise corresponds to the scenario of transport between source and drain. In this paper, it is shown that weak interactions lead to a correction of the localization length while a complete loss of coherence is observed for stronger interactions. I thus disagree with the author's conclusion that interactions do not play a role in their system. In fact, they miss the opportunity to study the interplay between interactions and Anderson localization following the lines of many previous experimental studies.

- 1.) We have conducted further numerical simulations, shown in the supplementary material, together with a new discussion in the main text, showing that the interaction at this level of dilution does not affect the transport properties.

2) I find it very surprising that the authors do not mention the value of the chemical potential of the condensate. In fact, the chemical potential and the associated healing length (ξ) is an important energy and length scale, respectively, never discussed in the text. It would allow evaluating how strong interactions are in their experiment. The only hint given is that they are in the Thomas-Fermi regime which points to the fact that interactions are non-negligible. I found it also surprising that the authors deem the thermal wavelength and associated momentum k_B more important than the momentum associated with the healing length $1/\xi$. In fact, the GPE theory they use to compare their experiment with is a $T=0$ theory and no thermal scales enter whatsoever. I, therefore, strongly suggest mentioning the value of the chemical potential and all other scales associated with it.

- 2.) We have added a calculation of the chemical potential and healing length to the supplementary material. We believe that the momentum distributions illustrated within the new Figure 3 provide a more relevant overview than the momentum associated with the healing length, as the momentum distribution is acquired from interaction-driven expansion out of the CO2 trap, following which the system approximates a single-particle experiment.

3) I disagree with the statement that the numerical simulations show quantitative agreement with the experiment. Especially for $\eta=0.32$ the agreement is at best qualitative. I also find it interesting how different the numerically calculated atomic densities look as compared to the absorption images in Fig. 2 a, b, c (why is the source so empty?). Unfortunately, the authors do not give any hints on the origin of these discrepancies. I also do not agree with the statement that the localization length in Fig. 2 d "approaches an asymptotic value for long localization times". In fact, the localization length slowly but steadily increases during the whole observation time again pointing at the importance of interactions.

- 3.) The experimental trapping potential is not precisely flat, and there is a weak fringing effect due to interference in the 2D trap. This is not modelled in the simulation, and accounts for minor discrepancies between theory and experiment. The period of any defects in the 2D trap is at least $10 \mu\text{m}$ and will not be a factor in the presence or absence of Anderson localisation. However, it will contribute to differences between experiment and theory, particularly in the case of zero disorder. Regarding the images in the simulation panel of Figure 2, we have smoothed the images with an $8 \mu\text{m}$ disk, consistent with the imaging resolution of our experimental setup. We note some differences with experiment concerning the filling of the source, but these do not affect the conclusions drawn regarding Anderson localisation.

4) I find the observation of a constant population in the channel quite interesting. At first sight, this might be very counterintuitive since naively one would expect that a larger localization length leads to a smaller population. However, it has been shown for scattering of linear waves in disorder that the dwell time is independent of the disorder strength, see e.g. Pierrat et al. PNAS 111, 17765 (2014). For linear waves the dwell time is associated with the fraction of the wavefunction inside the scattering region. One could directly compare the results of this paper with the so-called Weyl formula. As far as I know investigations of this effect including nonlinearity/interactions is lacking and would provide the current study with a further interesting physics question.

- 4.) We thank the referee for pointing us to Pierrat et al, which is very relevant to this work. We have incorporated the theory into the main text, relating the dwell time of the atoms in the channel to the equilibration time of the channel population.

Finally, a few more technical comments.

Main text:

In Fig.1 the image of the trap in the expanded bubble is very hard to see. Also the coordinate system does not seem to agree with the axis convention in all the following figures.

- 1.) We have changed the colour of Figure 1 and added some labels to aid interpretation. Additionally we have added panel (b) to improve the visualisation of the dynamics within the dumbbell. The coordinate system is correct, with atom flow along the x-direction, and z being the vertical (frozen) direction.

A plot of the trapping potential as a function of x (a cut through the potential) would be very helpful. At least it should appear in the supplement where the numerical simulations are discussed.

- 2.) We have added a cut of the potential to the supplementary material, in the new Figure 12.

The parameters (r,L,w) are not explained in the main text and one has to guess that it is the radius, the length, and the width of the trap.

- 3.) We have updated Figure 1 with the addition of subfigure (b), where we show the dumbbell and its dimension definitions clearly.

Why is the same averaging procedure as for the experimental data (see caption of Fig.2) not applied also to the numerical results?

- 4.) The same averaging procedure is also used for numerical simulation.

In the conclusions the sentence appears: "... and demonstrated that the transport may be tuned via disorder fill-factor from a regime of ballistic, to diffusive, and then to strongly localized transport". It should be made more clear that diffusive transport is associated with the period of time from which the resistance is taken.

- 5.) This sentence has been corrected as suggested.

The sentence in the conclusions "Our work demonstrated and explored the phenomenon..." reads like the repetition of the preceding sentence.

6.) The repetition has been corrected as suggested.

Supplementary material:

Overall, I find Figure 5 interesting/disturbing. While I agree that the randomly placed scatterers show nicer agreement to an exponential density profile, the regularly placed scatterers overall show the same behavior. The authors should explain the reason in more detail since it seems to hint to the excluded absence of classical percolation.

7.) Transport measurements and numerical simulation of the regular lattice have been added to the supplementary material.

The authors should give also the time step used in their numerical calculations since a standard Runge-Kutta-Fehlberg (please check spelling in your manuscript) for such systems is accurate only for sufficiently small time steps.

8.) The Runge-Kutta-Fehlberg technique is an adaptive method, and the time-step varies in order to maintain an error threshold of 7×10^{-12} in the L_2 norm of the wavefunction. We have added this information to the supplementary material, and corrected the spelling of Fehlberg.

In section E in the supplement reference 49 directs to the supplement and thus seems to be obsolete.

9.) The obsolete reference has been fixed as suggested

In the same section the unit of k_{dB} is given as m .

10.) The unit of k_{dB} has been corrected.

Overall I find Fig. 7 confusing:

- *In the main text it is mentioned that the scatterers are created via 2x2 pixels. Why is $w \geq 20px$ then relevant?*
- *Why is the data for $w=20px$ shown only for small fill-factors?*

11.) We have removed Figure 7 and its corresponding section from the supplementary material, as we find it to no longer be directly relevant to our discussion.

We emphasise that our manuscript shows comprehensive signatures for 2D AL and, in combination with system-realistic numerical modelling, rules out several other explanations, such as classical trapping and interaction-related phenomena. We again thank the referees for their insights which have improved our manuscript, and we ask for this manuscript to be reconsidered for publication in *Nature Communications*. We include all changes summarised below:

- Abstract: minor edits to short passages of text to improve the flow.
- Page 1, Paragraph 3: edit to 1st sentence to emphasise previous experiments in ^{87}Rb were with weakly interacting gas.
- Page 1, Paragraph 4: added reference 3 for exponential localisation.

- Page 2, Paragraph 2: added reference 45 for studies showing that localisation is maintained in the presence of interactions in 1D. Also corrected spelling of Runge-Kutta-Fehlberg and added statement highlighting the advantages of simulations for the study of the effect of interactions.
- Figure 1: added panel (b) and edited caption accordingly
- Page 3, Paragraph 2: Replaced reference to temperature with reference to wavenumber.
- Figure 2: smoothed simulation panels with 8 μm disk in accordance with experimental imaging resolution.
- Figure 3: added new Figure 3 which studies the k-vector distributions in all sections of the dumbbell and relates them to the expected localisation length from Kuhn et al.
- Page 4, Paragraph 2: Added reference 52 as a note, which states the difference in definitions of fill-factor between our work and the work of Morong and DeMarco.
- Page 4, Paragraph 5: removed reference to a Thomas-Fermi distribution as we provide fuller details on the k-vector distribution in Figure 3.
- Page 4, Paragraph 6: moved the sentence beginning with “We also confirm that the simulations exhibit an exponential channel profile with an asymptotic...” to the discussion in Page 5, Paragraph 2.
- Page 4, Paragraph 6: softened “quantitative agreement” to “reasonable quantitative agreement”.
- Page 5, Paragraph 2: New discussion on the differences between experiment and simulation.
- Page 5, Paragraphs 3 and 4: New discussion on the absence of classical trapping and the implications of the momentum distributions in Figure 3 and its link to previous theoretical work.
- Page 5, Paragraph 5: New discussion on the effect of interactions.
- Page 5, Paragraph 7: New interpretation excluding classical trapping: “We interpret this non-zero steady state number imbalance to be a consequence of a combination of Anderson localisation, and enhanced reflection into the source reservoir due to Anderson localisation.
- Page 5, Paragraph 8: Added discussion of the link to the channel dwell time to the equilibration time.
- Page 6, Paragraph 2: strengthened conclusion regarding the contribution to the resistance from high energy atoms based on the momentum distributions in Figure 3.
- Page 6, Paragraph 3: Added statement that the observation of exponential localisation cannot be explained by classical trapping within the channel.
- Page 6, Paragraph 3: Added statement describing the difference in equilibration time between simulation and experiment. Also removed redundant penultimate sentence.

Supplementary material:

- Section A, added paragraph with calculations of chemical potential and healing length.
- Section B and Figures 6-11: Added full experimental datasets for a range of different widths.
- Section C and Figure 12: Added cuts of the potential applied to the SLM.
- Figure 13: added numerical simulation panel for regular vs random lattice comparison.
- Section E: edited discussion based on Figure 13

- Section F: added discussion of the adaptive time-step in the Runge-Kutta-Fehlberg method.
- Section G and Figure 14: new section based on the effect of interactions.
- Section H: Based the discussion on the momentum distributions in Figure 3 instead of the temperature.
- Figure 15: replaced previous figure with simpler figure showing $\Lambda(|k|)$.
- Section “Length scale: average minimal distance” and the old Figure 7: removed from supplementary material as it is no longer directly relevant to the discussion.

REVIEWER COMMENTS

Reviewer #1 (Remarks to the Author):

I am now satisfied with the authors response to the comments in my first report. I recommend that this manuscript be published in Nature Communications.

Reviewer #2 (Remarks to the Author):

The authors have extensively revised the theoretical analysis and the discussion, following the suggestions by the referees.

In particular, they have performed new numerical simulations to investigate the potential effect of the finite interactions, to study the expected fraction of classically localized states, as well as the expected distribution of localization lengths. To my understanding, from these simulations they deduce that: a) The residual interactions in their very dilute samples is not likely to affect the dynamics. b) From the calculated momentum distributions and from the related theoretical prediction for the localization length, only a very small fraction of the sample is likely to undergo classical trapping, even at the largest fill factors. c) From the same calculations, it is instead likely that a relevant fraction of the sample has localization lengths longer than the channel width. This would justify the measurement of a finite resistance of the channel even in the regime in which the exponential profiles are observed. Employing the results of these new simulations, the authors have clarified some of my previous doubts, in particular points 1, 4, 6, and 8. In my opinion, this was an important improvement of the theoretical analysis, and the revised manuscript is more convincing about the possibility of observing Anderson localization in the experimental system.

The authors have also added to the supplementary material new experimental data on the channel profiles for various channel widths, to support the idea that the system has a true 2D nature or, in other words, that the channel width has not a big role in the dynamics. In response to my previous point 2, the authors state that the persistence of localization for a large range of widths (excluding the narrowest width) indicates that the environment cannot be 1D. They however recognize that the channel width might not be always larger than the mean localization length. This is something that would bring at least finite-size effects into the game, possibly changing the picture from the current theoretical understanding of the system. If I'm not wrong, the localization lengths shown in Fig.3 are indeed calculated in a truly 2D environment. Even the experimental localization lengths are typically larger than the channel widths, so it is probable that there are finite size effects at least in one of the two dimensions. From this point of view, it would be interesting to show also the measured localization length at a fixed filling factor (say 0.32) versus the channel width, for all the data shown in the supplementary material. If I try to guess from Figs. 7-11, the localization length is longer for narrower channels. If that is correct, one might ask whether this is an effect of the finite width, which perhaps tends to increase the effective localization length along the channel. Please note that I'm not going to ask the authors to provide all these analyses for the present paper. What I would like to note is instead that it is well known that the interesting aspect of Anderson localization in 2D is the energy-dependence of the localization length. The authors themselves note this in the

introduction: “localisation length in 2D depends exponentially on the particle energy [3, 34]: for experimentally feasible particle energies, observing localisation requires large systems (> 100 microns \times 100 microns) even for ultracold atoms.” Since the transverse size of their system is still below 100 microns, one would expect that finite size effects at some point will be taken into account.

Another aspect that seemed very interesting to me (my previous point 7) is the comparison between ordered and disordered scatterers, at an experimental level. My previous remark to the authors was: “I find this an important point, that might support the arguments in favour of Anderson localization.” Indeed, a qualitatively different behaviour of the system in the case of regular scatterers would have provided a very useful control experiment, excluding possible uncontrolled effects in the experiment system (such as the fringe problem mentioned by the authors). Unfortunately, the authors have not taken my suggestion to discuss this point more in depth.

Regarding instead some aspects of the transport experiment that were unclear to me (my previous point 3), the authors have performed an additional analysis based on the new ref. 56, and found that the coherent backscattering of atoms going from the source into the channel has an important role in the dynamics. That is very interesting, but I wonder whether the transport measurement is now even more disconnected from the profile analysis, at least for what regards the goal of demonstrating Anderson localization. In the conclusions, the authors continue to note that “The most conclusive experimental and numerical sign of the onset of Anderson localisation is the exponential channel profiles ...”

In conclusion, I think that both the experimental measurements and the theoretical analysis show very interesting phenomena related to Anderson localization in a 2D environment. The experimental geometry and the observable are very interesting, and the theoretical analysis is now convincing. There are however still some limitations of the present approach, such as the limited transverse size that will for sure impact the localization lengths. One important missing point, in my opinion, is the lack of a convincing control experiment that does show clearly a non-exponential profile in the absence of disorder, for example for a regular lattice. So, as the authors discuss clearly in the conclusions, the evidence for Anderson localization comes from an experiment-theory comparison. But it seems that there are still experimental imperfections that cannot be reliably modelled by the theory, such as the optical fringes, so the comparison is not fully quantitative.

Therefore, while from one side I would like to recommend publication of the present manuscript in Nature Communications because it presents very interesting, innovative work on localization in 2D disorder, from the other side I am very sceptical about the claim of having observed Anderson localization in 2D. I would like to note that the revised abstract does not claim anymore “Anderson localization” but only “exponential localization”. Also the reasoning in the conclusions on the interpretation of the exponential localization as Anderson localization is quite cautious. So, my suggestion is to scale down a bit the claim from “Anderson localization” to “exponential localization”, with a moderate revision of the discussion in the various parts of the manuscript. Alternatively, if dropping “Anderson localization” would not be possible, the authors should at least add a discussion on the possible impact of the limited transverse size of their system which, as I note above, contradicts their own definition of the minimum size that is necessary to observe Anderson

localization in 2D. In my opinion, a further revision of the manuscript in one or the other direction would not reduce the interest of the work but will probably spare the authors future criticisms, as unfortunately has happened already few times for experimental claims of Anderson localization. I think that the authors have a very interesting system at hand. After this initial demonstration of localization, they will for sure be able to carry out further work to study the fascinating properties predicted for Anderson localization in 2D.

Reviewer #3 (Remarks to the Author):

The authors have made significant changes to their paper as a response to the comments of all referees. While I appreciate the effort made, I would like to ask the authors to address also the points below before I can give my final approval.

One of my main points of criticism was on the role of interactions in the experiment. Even in case of very small interaction energies, which are much smaller than other energy scales, interactions may play an important role, however, on length and time scales not accessible in this experiment. I thus can follow the argument of the authors that in their experiment, the system might be indistinguishable from a non-interacting system. The main (numerical) underpinning for this statement is provided in Fig.14. in the supplement.

A few comments/questions/suggestions to this figure:

- Which parameters, especially which filling factor, have been used?
- Except for scattering length 0 for the last 30ms, none of the curves for the localization length shows real saturation. Since the authors claim that they see a saturated exponential channel profile only for $t > 400\text{ms}$ in their numerical results, why are the results shown only up to 250ms?
- Except for the very high factors of 6, 8, 10 as, there does not seem to be a systematic trend in the behavior of the curves as a function of the scattering length. Do the authors have at least a qualitative argument for this?
- I would also find it interesting to see the exponential density profile for the non-interacting case as compared to the case with full scattering length at times where saturation is reached. Could the authors provide such a figure?

As already mentioned, the authors claim now that the numerical simulations show a stationary density profile for longer times, i.e. $t > 400\text{ms}$ for both $\eta = 0.17$ and $\eta = 0.32$. Could the authors show the stationary density profiles obtained from the theory as compared to the stationary density profiles obtained from the experiment? For example, Fig.9 in the supplement would offer itself for such a comparison.

Regarding the momentum distributions, could the authors mention how they qualitatively change for the smaller filling factor of $\eta = 0.17$? Can one gain some intuition on the population imbalance from them in combination with the localization length as a function of k ? For example, for $\eta = 0.17$, I would guess that the momentum distribution in the drain gets overall broader as also particles with smaller k do not get localized. Is this correct? Since the integral over the momentum distribution in the channel remains essentially constant when varying η , how does the functional form change? To put it short, could the authors provide a more detailed discussion on the momentum distributions and their connection to the other observations.

In this context I do not understand the sentence in the manuscript: "This theoretical curve,...,

predicts localization lengths which are shorter than the system size for $k < 0.55 \mu\text{m}$, which agrees reasonably well with the in-channel momentum distribution in Fig.3(e) considering the exponential dependence." Fig.3(e) suggests that the majority of particles in the channel have localization lengths much larger than the channel length. These particles would then not be localized.

Finally, a few minor points:

The authors have modified the statement in the introduction as to the role of interactions in the 1D experiments. My comment, however, referred only to the Billy et al. experiment and not to the Roati et al. experiment where interactions have been tuned to zero via Feshbach resonances.

The authors claim that they have provided "the resistance data for the regular lattice". I did not find it.

We appreciate the positive feedback and constructive comments from the reviewers and are grateful for the opportunity to submit a revised manuscript accommodating the remaining suggestions. The additional results that we have added to Figs. 3 and 17, and the new figures Figs. 12, 13 and 14, further support our claim of having observed Anderson localisation in two dimensions. Following the previous suggestions of the reviewers, we have ruled out classical trapping, and additional numerical results provide significant insight into the role of interactions. We are convinced that the final manuscript will stimulate significant interest in the ultra-cold atom and solid-state communities and beyond.

We wish to note that in the course of revising the manuscript, we found an error in the processing of Figure 3 and Figure 18, which we have amended. The momentum distributions in Figure 3 now show the correct distributions for three different disorder levels. Figure 18 now scales $|k|$ correctly. These amended figures do not change our conclusions.

Below we include a point-by-point response to the referees' statements. We look forward to hearing back from you.

Kind regards

Donald White, Thomas Haase, Dylan Brown, Maarten Hoogerland, Mojdeh Najafabadi, John Helm, Christopher Gies, Daniel Schumayer and David Hutchinson

...

Reviewer #1 (Remarks to the Author):

*I am now satisfied with the authors response to the comments in my first report. **I recommend that this manuscript be published in Nature Communications.***

Reviewer #2 (Remarks to the Author):

The authors have extensively revised the theoretical analysis and the discussion, following the suggestions by the referees.

In particular, they have performed new numerical simulations to investigate the potential effect of the finite interactions, to study the expected fraction of classically localized states, as well as the expected distribution of localization lengths. To my understanding, from these simulations they deduce that: a) The residual interactions in their very dilute samples is not likely to affect the dynamics. b) From the calculated momentum distributions and from the related theoretical prediction for the localization length, only a very small fraction of the sample is likely to undergo classical trapping, even at the largest fill factors. c) From the same calculations, it is instead likely that a relevant fraction of the sample has localization lengths longer than the channel width. This would justify the measurement of a finite resistance of the channel even in the regime in which the exponential profiles are observed.

- Yes, these statements are correct.

*Employing the results of these new simulations, the authors have clarified some of my previous doubts, in particular points 1, 4, 6, and 8. In my opinion, this was an important improvement of the theoretical analysis, and the **revised manuscript is more convincing** about the possibility of observing Anderson localization in the experimental system.*

The authors have also added to the supplementary material new experimental data on the channel profiles for various channel widths, to support the idea that the system has a true 2D nature or, in other words, that the channel width has not a big role in the dynamics. In response to my previous point 2, the authors state that the persistence of localization for a large range of widths (excluding the narrowest width) indicates that the environment cannot be 1D. They however recognize that the channel width might not be always larger than the mean localization length. This is something that would bring at least finite-size effects into the game, possibly changing the picture from the current theoretical understanding of the system. If I'm not wrong, the localization lengths shown in Fig.3 are indeed calculated in a truly 2D environment. Even the experimental localization lengths are typically larger than the channel widths, so it is probable that there are finite size effects at least in one of the two dimensions. From this point of view, it would be interesting to show also the measured localization length at a fixed filling factor (say 0.32) versus the channel width, for all the data shown in the supplementary material. If I try to guess from Figs. 7-11, the localization length is longer for narrower channels. If that is correct, one might ask whether this is an effect of the finite width, which perhaps tends to increase the effective localization length along the channel. Please note that I'm not going to ask the authors to provide all these analyses for the present paper. What I would like to note is instead that it is well known that the interesting aspect of Anderson localization in 2D is the energy-dependence of the localization length. The authors themselves note this in the introduction: "localisation length in 2D depends exponentially on the particle energy [3, 34]: for experimentally feasible particle energies, observing localisation requires large systems (> 100 microns x 100 microns) even for ultracold atoms." Since the transverse size of their system is still below 100 microns, one would expect that finite size effects at some point will be taken into account.

- We appreciate the referee's concern regarding finite size effects in the system. In systems where the localisation length is significantly longer than either the channel width or the length, we agree that it may not be appropriate to interpret exponential localisation as implying Anderson localisation. Finite size effects may play a significant role in the data shown in the supplementary material below 50 μm channel widths, resulting in measured localisation lengths which are longer than the channel length and width. Above 50 μm widths, the system approximates a true 2D environment, and we do not observe a strong dependence of the localisation length on the width in the quasi-2D environment. We include the data above 50 μm widths in the new Figure 12 and Section C of the supplementary material. We exclude the data below 50 μm widths, because finite size effects appear to contribute to a lengthening of the localisation length. Exponential fitting for very long localisation lengths has a significant uncertainty, and the values obtained may mislead the reader. The purpose of Fig. 12 is to show that the data with 58 μm widths, as used in Figure 2 as the main evidence for Anderson localisation, has similar localisation properties for larger widths, indicating that finite size effects are not important for this data.

Another aspect that seemed very interesting to me (my previous point 7) is the comparison between ordered and disordered scatterers, at an experimental level. My previous remark to the authors was: "I find this an important point, that might support the arguments in favour of Anderson localization." Indeed, a qualitatively different behaviour of the system in the case of regular scatterers would have provided a very useful control experiment, excluding possible uncontrolled effects in the experiment system (such as the fringe problem mentioned by the authors). Unfortunately, the authors have not taken my suggestion to discuss this point more in depth.

Regarding instead some aspects of the transport experiment that were unclear to me (my previous point 3), the authors have performed an additional analysis based on the new ref. 56, and found that the coherent backscattering of atoms going from the source into the channel has an important role in the dynamics. That is very interesting, but I wonder whether the transport measurement is now even more disconnected from the profile analysis, at least for what regards the goal of demonstrating Anderson localization. In the conclusions, the authors continue to note that "The most conclusive experimental and numerical sign of the onset of Anderson localisation is the exponential channel profiles ..."

- We agree that the transport measurement, in and of itself, does not provide direct evidence of localisation. This is because the transport measurement mainly measures weakly localised atoms which have sufficient energy to traverse the channel. The updated analysis in Fig. 3 shows that the disordered channel acts as an effective high-pass energy filter. However, these channel resistance measurements are still a valuable tool for describing the system, as they permit an analysis of the behaviour of weakly localised atoms, and the analysis in Fig. 5 shows that the resistance increases with increasing fill-factor.

*In conclusion, I think that both the experimental measurements and the theoretical analysis show very interesting phenomena related to Anderson localization in a 2D environment. **The experimental geometry and the observable are very interesting, and the theoretical analysis is now convincing.** There are however still some limitations of the present approach, such as the limited transverse size that will for sure impact the localization lengths. One important missing point, in my opinion, is the lack of a convincing control experiment that does show clearly a non-exponential profile in the absence of disorder, for example for a regular lattice.*

- From our point of view, the control experiments are taken to be the weak fill-factor data, where we do not observe long-term exponential density profiles in the channel. The regular lattice experiment and numerical simulation in the supplementary material shows a difference between order and disorder. However, the disordered data in this comparison is not in a regime of strong localisation ($\eta = 0.13$). To go beyond our theoretical investigation, we have planned an experimental comparison between order and disorder for future work.

So, as the authors discuss clearly in the conclusions, the evidence for Anderson localization comes from an experiment-theory comparison. But it seems that there are still experimental imperfections that cannot be reliably modelled by the theory, such as the optical fringes, so the comparison is not fully quantitative.

- Yes, this is true, and it is one of the main reasons that an experimental setup is a crucial part of any quantum simulator of Anderson localisation. The theoretical studies in this manuscript support our experimental observations to a level of reasonable quantitative agreement. Any experimental imperfections, which we do not theoretically model, do not affect the qualitative agreement we observe between experiment and theory. At the same time, we would like to stress that the numerical simulations allow us to provide additional insight beyond experimental possibilities, such as to shine light on the role of interaction effects, and of the momentum distributions in different spatial regions, which is a great benefit of our cooperative approach.

Therefore, while from one side I would like to recommend publication of the present manuscript in Nature Communications because it presents very interesting, innovative work on localization in 2D disorder, from the other side I am very sceptical about the claim of having observed Anderson localization in 2D. I would like to note that the revised abstract does not claim anymore "Anderson localization" but only "exponential localization". Also the reasoning in the conclusions on the interpretation of the exponential localization as Anderson localization is quite cautious. So, my suggestion is to scale down a bit the claim from "Anderson localization" to "exponential localization", with a moderate revision of the discussion in the various parts of the manuscript. Alternatively, if dropping "Anderson localization" would not be possible, the authors should at least add a discussion on the possible impact of the limited transverse size of their system which, as I note above, contradicts their own definition of the minimum size that is necessary to observe Anderson localization in 2D.

- We have clarified our conclusions to state that we observe exponential localisation, and that our interpretation of this exponential localisation is that Anderson localisation is present in the system when the localisation length is shorter than the system size. We have provided complementary analyses of both the measurements and simulations, which both support the interpretation that we have observed Anderson localisation in two dimensions. Our analysis largely rules out the competing hypothesis of classical trapping, and we thus conclude that the system exhibits exponential Anderson localisation of low energy atoms.

*In my opinion, a further revision of the manuscript in one or the other direction would not reduce the interest of the work but will probably spare the authors future criticisms, as unfortunately has happened already few times for experimental claims of Anderson localization. **I think that the authors have a very interesting system at hand. After this initial demonstration of localization, they will for sure able to carry out further work to study the fascinating properties predicted for Anderson localization in 2D.***

Reviewer #3 (Remarks to the Author):

The authors have made significant changes to their paper as a response to the comments of all referees. While I appreciate the effort made, I would like to ask the authors to address also the points below before I can give my final approval.

One of my main points of criticism was on the role of interactions in the experiment. Even in case of very small interaction energies, which are much smaller than other energy scales, interactions may

play an important role, however, on length and time scales not accessible in this experiment. I thus can follow the argument of the authors that in their experiment, the system might be indistinguishable from a non-interacting system. The main (numerical) underpinning for this statement is provided in Fig.14. in the supplement.

A few comments/questions/suggestions to this figure:

- Which parameters, especially which filling factor, have been used?
 - In the previous version of the manuscript, we used a filling factor of 0.17 in this figure. In the new edition of the manuscript, we have changed this to 0.32, as it shows stronger localisation. We have clarified the parameters in the caption.
- Except for scattering length 0 for the last 30ms, none of the curves for the localization length shows real saturation. Since the authors claim that they see a saturated exponential channel profile only for $t > 400\text{ms}$ in their numerical results, why are the results shown only up to 250ms?
 - In the new edition of the manuscript, we have extended the time shown to 750ms. Previously, we showed the time up to 250ms because it avoids complications associated with atoms from the drain reservoir re-entering the channel, which has the effect of appearing to lengthen the localisation length, in agreement with numerical simulation. We have now included data up to 750 ms along with a statement in Section J of the supplementary material stating that any slow increase of the apparent localisation length should not necessarily be viewed as a weakening of localisation.
- Except for the very high factors of 6, 8, 10 as, there does not seem to be a systematic trend in the behavior of the curves as a function of the scattering length. Do the authors have at least a qualitative argument for this?
 - Our argument is that this is essentially a single-particle experiment. Very low interaction strengths do not have a significant effect on the dynamics.
- I would also find it interesting to see the exponential density profile for the non-interacting case as compared to the case with full scattering length at times where saturation is reached. Could the authors provide such a figure?

- Above, we include the simulated density profile with zero interaction and with the interaction of Rb, in the case of $\eta = 0.32$, $(r, L, w) = (43, 108, 58) \mu\text{m}$, and an expansion time of 350 ms. We do not observe a significant difference with and without interaction.

As already mentioned, the authors claim now that the numerical simulations show a stationary density profile for longer times, i.e. $t > 400\text{ms}$ for both $\eta = 0.17$ and $\eta = 0.32$. Could the authors show the stationary density profiles obtained from the theory as compared to the stationary density profiles obtained from the experiment? For example, Fig.9 in the supplement would offer itself for such a comparison.

- Long-time density profiles from numerical simulation are now shown in Figure 13. The figure shows that the density profile in the case of $\eta = 0.07$ evolves to a non-exponential state (note in particular the increasing density from $x = 0 \rightarrow 75 \mu\text{m}$). In contrast, the exponential character is maintained for $\eta = 0.17$ and $\eta = 0.32$.

Regarding the momentum distributions, could the authors mention how they qualitatively change for the smaller filling factor of $\eta = 0.17$? Can one gain some intuition on the population imbalance from them in combination with the localization length as a function of k ? For example, for $\eta = 0.17$, I would guess that the momentum distribution in the drain gets overall broader as also particles with smaller k do not get localized. Is this correct? Since the integral over the momentum distribution in the channel remains essentially constant when varying η , how does the functional form change? To put it short, could the authors provide a more detailed discussion on the momentum distributions and their connection to the other observations.

-We appreciate this comment from the referee, and we believe that the momentum distributions provide valuable insight. As a result, we have also included the momentum distributions in all three regions of the dumbbell for fill-factors 0.00 and 0.17 in Figure 3. We have added a discussion to Page 5, paragraph 4 specifically regarding the momentum distribution within the drain, showing that the channel acts as a high-pass energy filter. With increasing disorder, we observe that more low-energy atoms are present in the channel.

In this context I do not understand the sentence in the manuscript: "This theoretical curve,..., predicts localization lengths which are shorter than the system size for $k < 0.55 \mu\text{m}^{-1}$, which agrees reasonably well with the in-channel momentum distribution in Fig.3(e) considering the exponential dependence." Fig.3(e) suggests that the majority of particles in the channel have localization lengths much larger than the channel length. These particles would then not be localized.

- Our argument is that the theoretical prediction of Equation 3 in the manuscript is a rather general statement, and is not specialised to the specifics of our system; for example, it does not consider the finite size of our system or the unique nature of point disorder. As a result, the calculated value of $0.55 \mu\text{m}^{-1}$ should be regarded as a ballpark figure, which agrees relatively well with the peak value of the atom momenta within the channel. We have also included a statement regarding the previous findings of Morong and DeMarco, which showed a sub-exponential dependence of localisation length on energy in point disorder; we do not make any comment on the validity of their findings,

except to say that Equation (3) of our manuscript may not encapsulate the full physics of a system with point disorder.

Finally, a few minor points:

The authors have modified the statement in the introduction as to the role of interactions in the 1D experiments. My comment, however, referred only to the Billy et al. experiment and not to the Roati et al. experiment where interactions have been tuned to zero via Feshbach resonances.

- We have clarified this in the manuscript.

The authors claim that they have provided “the resistance data for the regular lattice”. I did not find it.

- We have added the numerical values for the resistance to the discussion in Section H of the supplementary material.

Below we list all changes made to the manuscript:

- Page 1, Paragraph 3: distinction in references between weakly- and non-interacting gases
- Page 2, Paragraph 1: addition of text: “...and observe **compelling evidence for Anderson localisation...**”
- Page 3, Paragraph 2: correction of mean wavenumber from $0.9 \mu\text{m}^{-1}$ to $1.6 \mu\text{m}^{-1}$
- Page 4, Paragraph 3: “...which allows direct observation of the presence of Anderson localisation...” is changed to “...which allows direct observation of exponential localisation...”
- Page 4, Paragraph 5: “...approaches an asymptotic value for long times.” is changed to “...approaches a quasi-stationary value for long times.”
- Page 5, Paragraph 1: end of paragraph contains a statement regarding finite size effects.
- Page 5, Paragraph 2: addition of text: “...do tend to a **near-flat** constant density profile.” with reference to supplementary material.
- Page 5, Paragraph 2: “...with an asymptotic mean localisation length...” is changed to “...with a quasi-stationary mean localisation length...”
- Page 5, Paragraph 3: modification of paragraph to solidify the conclusions. Also removed statements regarding phase noise simulations as they are now incidental to this work.
- Figure 3: corrected error in previous manuscript regarding data analysis, affecting Figures (a), (b), (d), (e) and (f). Added fill-factors 0.00 and 0.17 to (d), (e) and (f). Also made corresponding edits to caption.
- Page 5, Paragraph 4: modification of paragraph to discuss in detail the momentum distributions in Figure 3.
- Page 5, Paragraph 5: modification of discussion regarding Figure 3(c), and inclusion of reference to Morong and DeMarco’s findings. Also made Equation 3 into a standalone equation.

- Page 5, Paragraph 6: “...to significantly affect the observation of Anderson localisation.” changed to “...to significantly alter the observed density profiles.”
- Page 6, Paragraph 1: “...although strong Anderson localisation is not observed...” changed to “...although exponential localisation is not observed...”
- Figure 5 caption: details of dumbbell radius and width added
- Page 6, Paragraph 4: modification of conclusion to clarify that we have observed compelling evidence for Anderson localisation, and that our interpretation of the data is that exponential localisation implies Anderson localisation for $\xi < L$.

Supplementary material:

- Figures 6-11: Captions modified to include “This data is an average of three experimental disorder realisations.”
- Figure 12: new figure showing width dependence in quasi-2D environment
- Section C: new section discussing Fig. 12
- Figure 13: new figure showing long-time simulations
- Section D: new section discussing Fig. 13
- Figure 14: new figure showing simulated momentum distributions for 3 different fill-factors.
- Section E: new section discussing Fig. 14
- Figure 16: caption amended to include dumbbell radius, length and width
- Section H: numerical values of resistance added for disorder and regular lattice.
- Figure 17: Figure amended to include data for longer times up to 750 ms. This data now shows simulations for $\eta = 0.32$ data due to stronger localisation. The caption has been amended accordingly.
- Section I: Paragraph beginning “We note that the slow apparent increase in localisation length...” added.
- Section K: “...we emphasise that there is a Thomas-Fermi distribution...” changed to “we emphasise that there is a distribution...”
- Equation 5: error in equation corrected (square-root was incorrect), and the value of the de Broglie wavelength has been amended to match the distribution in Fig. 3(b).
- Section K: Following “...their relationship in the sequence of our experimental runs is ...” the order of length-scales has been modified to include the localisation length and to rearrange the de Broglie wavelength to be shorter than ℓ_{tr} .
- Section K: “The random scatterers have a strength of...” changed to “The random scatterers have a height of...”
- Section K: “Using the approximation $k \approx k_{dB}$, one obtains $k_{dB}\sigma \approx 2.2$ ”, correcting the previous value of 1.0.
- Figure 18: this figure has been amended to scale the horizontal axis correctly; previously, the horizontal axis was mistakenly plotted as $|k|\sigma$ instead of $|k|$.

REVIEWERS' COMMENTS

Reviewer #2 (Remarks to the Author):

The authors have further improved the presentation and discussion of the data, in response to the criticisms by two of the referees. I think that now the manuscript presents a satisfying account of the experimental data, of the numerical simulations and of their comparison with the established theory of Anderson localization.

As I already said, this work shows very interesting phenomena related to Anderson localization in a 2D environment, using an innovative setup. I am sure that that these results will be of interest of a wide community. However, establishing unambiguously Anderson localization in 2D by assessing the peculiar properties of e.g. the localization length vs energy, as recalled in the introduction, will require more work. In my opinion, the manuscript still carries traces of the correct doubts by the same authors about an “unambiguous observation of Anderson localisation in 2D”. The abstract indeed claims just “exponential localisation in a 2D ultracold atom system”, and the conclusions are also quite cautious: “We therefore interpret profiles with localisation lengths shorter than the channel length to signify Anderson localisation in 2D. ... The simulations provide additional insight into the role of interactions and the momentum distributions at different fill factors, corroborating the experimental evidence, and providing strong support that Anderson localisation is the suitable interpretation of the exponential density profiles and of the reduced transport”.

So, in giving my recommendation towards publication of the manuscript in Nature Communications, I would once more suggest considering whether a more cautious title (e.g. “Probable observation ...”) would better reflect the contents of the paper.

Please note the final sentence “We note that this is the first observation known to us of Anderson localisation in 2D ultracold atom systems” is a repetition of a similar sentence on page 1, beginning of the column 2. So, I suggest removing it.

Reviewer #3 (Remarks to the Author):

The authors have made again significant changes to their paper to address the concerns and questions of all referees. I now agree with the main conclusions of the paper and recommend it for publication in Nature Communications.

We thank the referees for their constructive feedback throughout this review process, and we thank them for their recommendations to publish our manuscript as an article in Nature Communications. Below we detail our response to their final points.

REVIEWERS' COMMENTS

Reviewer #2 (Remarks to the Author):

The authors have further improved the presentation and discussion of the data, in response to the criticisms by two of the referees. **I think that now the manuscript presents a satisfying account of the experimental data, of the numerical simulations and of their comparison with the established theory of Anderson localization.**

As I already said, this work shows very interesting phenomena related to Anderson localization in a 2D environment, using an innovative setup. I am sure that that these results will be of interest of a wide community. However, establishing unambiguously Anderson localization in 2D by assessing the peculiar properties of e.g. the localization length vs energy, as recalled in the introduction, will require more work. In my opinion, the manuscript still carries traces of the correct doubts by the same authors about an “unambiguous observation of Anderson localisation in 2D”. The abstract indeed claims just “exponential localisation in a 2D ultracold atom system”, and the conclusions are also quite cautious: “We therefore interpret profiles with localisation lengths shorter than the channel length to signify Anderson localisation in 2D. ... The simulations provide additional insight into the role of interactions and the momentum distributions at different fill factors, corroborating the experimental evidence, and providing strong support that Anderson localisation is the suitable interpretation of the exponential density profiles and of the reduced transport”.

- We appreciate the referee’s adherence to scientific rigour. We are clear in our conclusions that we have observed exponential localisation, and that we can find no competing theories to explain these observations. The conditions fit the established theoretical limits for Anderson localisation in 2D. Therefore, our interpretation of the results is the observation of Anderson localisation in 2D.

So, in giving my recommendation towards publication of the manuscript in Nature Communications, I would once more suggest considering whether a more cautious title (e.g. “Probable observation ...”) would better reflect the contents of the paper.

- We respectfully disagree with the referee’s suggestion to edit the title. In the spirit of empirical falsification, we have proposed competing theories explaining our observations (namely, classical trapping), and have shown that they do not support our observations. Moreover, we have incorporated relevant theoretical studies to show that our system conditions are within the bounds under which Anderson localization can be expected to be observed in two dimensions. Within the bounds of existing theoretical knowledge of Anderson localization, our position is that we have observed the phenomenon. We also note that two out of three referees agree with all conclusions and with the title, and that Referee 2 only ‘suggests considering’ an edit to the title. While we agree with the importance of doubt (and the impossibility of proof) within

science, we believe that our observations should stand as they are, and we should allow the scientific community to analyse and debate them as they stand.

Please note the final sentence “We note that this is the first observation known to us of Anderson localisation in 2D ultracold atom systems” is a repetition of a similar sentence on page 1, beginning of the column 2. So, I suggest removing it.

- We have removed the sentence.

Reviewer #3 (Remarks to the Author):

The authors have made again significant changes to their paper to address the concerns and questions of all referees. **I now agree with the main conclusions of the paper and recommend it for publication in Nature Communications.**

List of Changes

- We have modified the abstract in accordance with the specified format.
- We have updated all figures to include mathematical symbols in italics.
- We have updated the text to Romanise subscripts.
- We have divided the manuscript into sections, and the Results section into subsections.
- We have moved the two paragraphs beginning with “Our experimental observations are complemented...” to the “Theory” subsection of the Results section.
- In keeping with the Nature journals’ custom of placing the Methods section at the end of the article, we have moved the description of Fig. 1 to the beginning of the “Evolution of Channel Density Profiles” Results subsection.
- We have included descriptions of the errorbars in all figures.
- The axes limits of Figure 5 have been slightly modified in order to begin at $L = 0$.
- We have moved the single footnote into the main text (the final sentence of the Methods section).
- We have removed the sentence suggested by Referee 2.
- We have modified the first sentence of the final paragraph for better flow.
- We have added an acknowledgement for S. S. Shamailov.
- We have modified the Author Contributions section to include D. H. W. as contributing to the data analysis.
- We have removed the word “financial” from the competing interests section.